# Single-cell analysis uncovers that metabolic reprogramming by ErbB2 signaling is essential for cardiomyocyte proliferation in the regenerating heart

Hessel Honkoop[1†], Dennis EM de Bakker[1†], Alla Aharonov[2], Fabian Kruse[1], Avraham Shakked[2], Phong D Nguyen[1], Cecilia de Heus[3], Laurence Garric[1], Mauro J Muraro[1], Adam Shoffner[4], Federico Tessadori[1], Joshua Craiger Peterson[5], Wendy Noort[5], Alberto Bertozzi[6], Gilbert Weidinger[6], George Posthuma[3], Dominic Grün[7], Willem J van der Laarse[8], Judith Klumperman[3], Richard T Jaspers[5], Kenneth D Poss[4], Alexander van Oudenaarden[1], Eldad Tzahor[2], Jeroen Bakkers[1,9]*

[1]Hubrecht Institute-KNAW and University Medical Center Utrecht, Utrecht, Netherlands; [2]Department of Molecular Cell Biology, Weizmann Institute of Science, Rehovot, Israel; [3]Section Cell Biology, Center for Molecular Medicine, University Medical Center Utrecht, Utrecht University, Utrecht, Netherlands; [4]Regeneration Next, Department of Cell Biology, Duke University Medical Center, Durham, United States; [5]Laboratory for Myology, Department of Human Movement Sciences, Faculty of Behavioural and Movement Sciences, Vrije Universiteit Amsterdam, Amsterdam, Netherlands; [6]Institute of Biochemistry and Molecular Biology, Ulm University, Ulm, Germany; [7]Max Planck Institute of Immunobiology and Epigenetics, Freiburg, Germany; [8]Department of Physiology, Institute for Cardiovascular Research, VU University Medical Center, Amsterdam, Netherlands; [9]Department of Medical Physiology, Division of Heart and Lungs, University Medical Center Utrecht, Utrecht, Netherlands

*For correspondence:
j.bakkers@hubrecht.eu

†These authors contributed equally to this work

Competing interests: The authors declare that no competing interests exist.

**Abstract** While the heart regenerates poorly in mammals, efficient heart regeneration occurs in zebrafish. Studies in zebrafish have resulted in a model in which preexisting cardiomyocytes dedifferentiate and reinitiate proliferation to replace the lost myocardium. To identify which processes occur in proliferating cardiomyocytes we have used a single-cell RNA-sequencing approach. We uncovered that proliferating border zone cardiomyocytes have very distinct transcriptomes compared to the nonproliferating remote cardiomyocytes and that they resemble embryonic cardiomyocytes. Moreover, these cells have reduced expression of mitochondrial genes and reduced mitochondrial activity, while glycolysis gene expression and glucose uptake are increased, indicative for metabolic reprogramming. Furthermore, we find that the metabolic reprogramming of border zone cardiomyocytes is induced by Nrg1/ErbB2 signaling and is important for their proliferation. This mechanism is conserved in murine hearts in which cardiomyocyte proliferation is induced by activating ErbB2 signaling. Together these results demonstrate that glycolysis regulates cardiomyocyte proliferation during heart regeneration.

**eLife digest** Heart attacks are a common cause of death in the Western world. During a heart attack, oxygen levels in the affected part of the heart decrease, which causes heart muscle cells to die. In humans the dead cells are replaced by a permanent scar that stabilizes the injury but does not completely heal it. As a result, individuals have a lower quality of life after a heart attack and are more likely to die from a subsequent attack.

Unlike humans, zebrafish are able to regenerate their hearts after injury: heart muscle cells close to a wound divide to produce new cells that slowly replace the scar tissue and restore normal function to the area. It remains unclear, however, what stimulates the heart muscle cells of zebrafish to start dividing. To address this question, Honkoop, de Bakker et al. used a technique called single-cell sequencing to study heart muscle cells in wounded zebrafish hearts.

The experiments identified a group of heart muscle cells close to the site of the wound that multiplied to repair the damage. This group of cells had altered their metabolism compared to other heart muscle cells so that they relied on a pathway called glycolysis to produce the energy and building blocks they needed to proliferate. Blocking glycolysis impaired the ability of the heart muscle cells to divide, indicating that this switch is necessary for the heart to regenerate. Further experiments showed that a signaling cascade, which includes the molecules Nrg1 and ErbB2, induces heart muscle cells in both zebrafish and mouse hearts to switch to glycolysis and undergo division.

These findings indicate that activating glycolysis in heart muscle cells may help to stimulate the heart to regenerate after a heart attack or other injury. The next step following on from this work is to develop methods to activate glycolysis and promote cell division in injured hearts.

## Introduction

Within the animal kingdom, regenerative capacity of damaged organs and body parts differs widely. While regenerative capacity is generally low in mammalian species, this can be very efficient in specific fish and amphibians (*Poss, 2010*). Even amongst animals of the same species, such as Astyanax mexicanus a teleost fish like the zebrafish, regenerative capacity can vary significantly (*Stockdale et al., 2018*). A better understanding of these differences and the cellular and molecular processes during tissue regeneration might ultimately help to improve the regenerative capacity of organs and tissues with poor intrinsic regenerative capacity.

The mammalian adult heart has very little regenerative capacity. The myocardium lost after an injury is replaced by scar tissue, which does not contribute to cardiac contraction resulting in reduced pumping efficiency and ultimately heart failure. Although a low level of cardiomyocyte turnover has been observed, there is no evidence of a stem cell population in the mammalian heart (*Kretzschmar et al., 2018*; *van Berlo et al., 2014*). Instead, a rare population of endogenous cardiomyocytes retains the capacity to proliferate to maintain cardiac homeostasis (*Bergmann et al., 2009*; *Bergmann et al., 2015*; *Kimura et al., 2015*; *Senyo et al., 2013*). Contrary to the adult heart, the neonatal mouse and potentially human heart still have the capacity to regenerate after injury (*Haubner et al., 2016*; *Porrello et al., 2011*). This regenerative capacity of the neonatal heart involves proliferation of pre-existing cardiomyocytes and is lost soon after birth most likely due to a sharp decrease in cardiomyocyte proliferation (*Porrello et al., 2011*; *Soonpaa et al., 1996*). Recent efforts to enhance proliferation of cardiomyocytes by either inhibiting Hippo signaling or activating the Nrg1/ErbB2 signaling pathway in the adult mammalian heart have been successful and show the potential of existing cardiomyocytes to reenter the cell cycle. (*Bersell et al., 2009*; *D'Uva et al., 2015*; *Heallen et al., 2011*).

The zebrafish heart regenerates very efficiently and will regrow cardiac muscle without scarring (*Poss et al., 2002*). Lineage tracing experiments revealed a model in which proliferation of preexisting cardiomyocytes replaces the myocardium that was lost during the injury (*Jopling et al., 2010*; *Kikuchi et al., 2010*). While cardiomyocyte proliferation in the uninjured adult zebrafish heart is very low, 3 days after the injury (dpi) cardiomyocytes near the injury area (so-called border zone) start to reenter the cell cycle as observed by the induction PCNA expression, phosphorylated histone three and BrdU incorporation and proliferation peaks at 7 days post injury (dpi) (*Jopling et al., 2010*;

*Kikuchi et al., 2010*; *Chablais et al., 2011*; *González-Rosa et al., 2011*; *Schnabel et al., 2011*; *Wu et al., 2016*). At the time when proliferation is observed, cardiomyocytes located in the border zone start to express cardiac transcription factors known for their role in embryonic heart development such as *nkx2.5* and *tbx20* and activate *gata4* enhancer elements (*Kikuchi et al., 2010*; *Lepilina et al., 2006*). In addition, border zone cardiomyocytes show signs of dedifferentiation such as disorganization of sarcomere structures and the reexpression of embryonic myosins (*Jopling et al., 2010*; *Wu et al., 2016*). There is increasing evidence that other (non-muscle) cells in the heart secrete growth factors that stimulate cardiomyocyte proliferation including retinoic acid, TGF-b ligands, insulin-like growth factor, Hedgehog, and Neuregulin (*Chablais and Jazwinska, 2012*; *Choi et al., 2013*; *Dogra et al., 2017*; *Gemberling et al., 2015*; *Lepilina et al., 2006*; *Wu et al., 2016*; *Zhao et al., 2019*; *Zhao et al., 2014*). In addition to these growth factors, prolonged hypoxia stimulates cardiomyocyte proliferation (*Jopling et al., 2012*; *Marques et al., 2008*).

The proliferating cardiomyocytes exist within a heterogeneous cell population including non-proliferating cardiomyocytes, endothelial cells and immune cells, hampering the discovery of genetic programs specific for these proliferating cardiomyocytes using whole tissue or spatially resolved RNA-sequencing (RNA-seq) approaches (*Kang et al., 2016*; *Lien et al., 2006*; *Sleep et al., 2010*). To identify molecular processes that differ between proliferating and non-proliferating cardiomyocytes, we explored a single-cell RNA-seq approach using the regenerating zebrafish heart. We found that upon injury, adult border zone cardiomyocytes dedifferentiate and resemble embryonic cardiomyocytes on a transcriptomic level. In addition, while adult cardiomyocytes mainly rely on fatty acid metabolism and mitochondrial oxidative phosphorylation (OXPHOS), border zone cardiomyocytes have reduced mitochondrial OXPHOS activity while genes encoding enzymes for glycolysis are induced and glucose uptake is enhanced. Importantly, Nrg1/ErbB2 signaling is sufficient to induce metabolic reprogramming in adult cardiomyocytes of both zebrafish as well as the murine hearts. In addition, the metabolic reprogramming from mitochondrial OXPHOS to glycolysis is required for efficient cardiomyocyte proliferation.

Together, these data support a model in which cardiomyocytes located in the border zone of the regenerating zebrafish heart undergo metabolic reprogramming, which is essential for cardiomyocyte proliferation and that this mechanism is conserved in a murine model with Nrg1/ErbB2 induced regeneration.

## Results

### Single-cell RNA-seq reveals transcriptionally distinct border zone cardiomyocytes

The border zone comprises only a small fraction of the total number of cardiomyocytes in the injured ventricle (*Wu et al., 2016*). Several genes and regulatory sequencing have been identified that mark border zone cardiomyocytes, including *nppa*, which encodes for the cardiac stress hormone ANF (*Kikuchi et al., 2010*; *Wu et al., 2016*). To mark these borderzone cardiomyocytes we generated a transgenic zebrafish *nppa* reporter line (*TgBAC(nppa:mCitrine)*) in which mCitrine expression recapitulates endogenous cardiac expression of *nppa* (*Figure 1—figure supplement 1a–e*). While low *nppa:mCitrine* expression was observed in trabecular cardiomyocytes of the remote area, higher expression was detected in the trabecular and cortical cardiomyocytes close to the injured area (*Figure 1a* and *Figure 1—figure supplement 1e*). Moreover, expression of *nppa* correlates with previously reported border zone activity of *gata4* regulatory elements (*Figure 1—figure supplement 1f*) (*Kikuchi et al., 2010*). Histochemical analysis of cryo-injured adult hearts revealed that 75% (±7%, n = 3) of the cardiomyocytes expressing high levels of *nppa:mCitrine* reentered the cell cycle (*Figure 1a*). To obtain border zone (proliferating) and remote (non-proliferating) cardiomyocytes from the same tissue for further analysis, we cryo-injured *nppa:mCitrine* hearts followed by cell dissociation and FACS sorting for both mCitrine[high] and mCitrine[low] cells (*Figure 1b*). Individual, living cells were sorted, followed by single-cell mRNA-sequencing using the SORT-seq (SOrting and Robot-assisted Transcriptome SEQuencing) platform (*Muraro et al., 2016*) (*Figure 1—source data 1*). In total 768 cells where sequenced in which we detected 19257 genes. We detected an average of 10,443 reads per cell and we introduced a cutoff at minimally 3500 reads per cell before further analysis, which resulted in the analysis of 352 cells. To identify the cardiomyocytes amongst the other

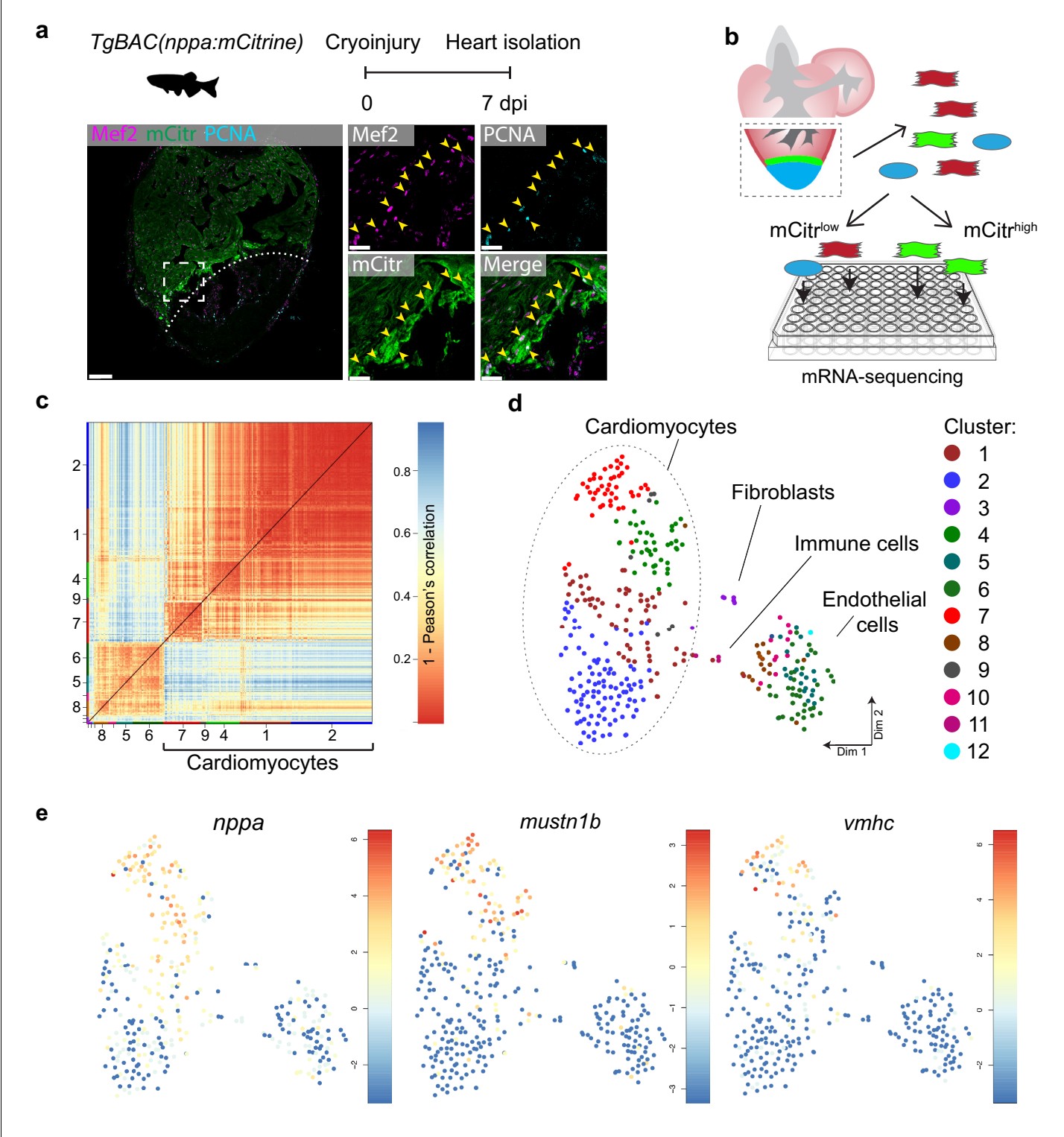

**Figure 1.** Single-cell mRNA sequencing identifies different cardiomyocyte-populations in the injured zebrafish heart. (a) Schematic of cryoinjury procedure on adult TgBAC(*nppa*:mCitrine) fish and immunohistochemistry on section of injured *TgBAC(nppa:mCitrine)* heart 7 dpi. Overview image on left and zoom-in of boxed region on the right. Mef2 (in magenta) labels cardiomyocytes, nppa:mCitrine (in green) marks the borderr zone, and PCNA (in cyan) marks proliferating cells. Arrows indicate triple-positive cells. Dashed line indicates injury site. Scale bar in overview 50 μm. Scale bar in zoom-ins 20 μm. (b) Experimental outline of the single-cell mRNA-sequencing of injured zebrafish hearts (blue, injury area; green, border zone) (c) Pairwise correlation between individual cells across all genes detected. Color-code indicates cell-to-cell distances measured by [1 – Pearson's correlation

*Figure 1 continued on next page*

*Figure 1 continued*

coefficient]. StemID clusters are indicated by color and number on the x- and y-axis. (d) t-distributed stochastic neighbor embedding (tSNE) map representation of transcriptome similarities between individual cells. (e) tSNE maps visualizing log2-transformed read-counts of the border zone marker genes *nppa*, *mustn1b* and *vmhc*.

The online version of this article includes the following source data and figure supplement(s) for figure 1:

**Source data 1.** Single-cell mRNA sequencing data from cryoinjured *TgBAC(nppa:mCitrine)* zebrafish hearts at 7 dpi.
**Source data 2.** List of the clusters in the adult injured heart (raw data see SD1), as identified by StemID and the associated cells per cluster.
**Source data 3.** List of differentially expressed genes between cardiomyocytes clusters 2 and 7 of the adult heart dataset.
**Figure supplement 1.** *TgBAC(nppa:mCitrine)* expression recapitulates endogenous *nppa* gene expression.
**Figure supplement 2.** Single-cell mRNA sequencing identifies different cell-populations in the injured zebrafish heart.
**Figure supplement 3.** Cluster 7 cells display highest nppa:mCitrine expression.
**Figure supplement 4.** Enhanced expression of genes elevated in cluster 7 versus cluster 2 is injury induced.

cell types, we first identified the different cell types based on their transcriptomes. k-medoids clustering of the single cell transcriptomes by the RaceID clustering algorithm was used (*Grün et al., 2015*) (*Figure 1c* and *Figure 1—source data 2*), and visualized in two dimensions using *t*-distributed stochastic neighbor embedding (*t*-SNE) (*Figure 1d*). A total of 12 cell clusters were identified, including a large group of cardiomyocytes (clusters 1,2,4,7 and 9), a smaller group of endothelial cells (clusters 5,6,8,10 and 12), and some fibroblasts (cluster 3) and immune cells (cluster 11) using the expression of marker genes for specific cell types (*Figure 1—figure supplement 2*). Based on the transcriptome clustering, the cardiomyocytes fell into four main transcriptionally-defined clusters (1, 2, 4 and 7), indicating that the injured heart contained subgroups of cardiomyocytes. To address whether the border zone cardiomyocytes were enriched in one of the four cardiomyocyte clusters we compared the mCitrine fluorescence intensity (recorded during FACS sorting) of the cardiomyocyte and found that the average intensity was highest in cluster 7 and lowest in cluster 2 (*Figure 1—figure supplement 3*). In addition, we analysed the single-cell transcriptome data for the expression of *nppa* and compared this to the expression of *vmhc* and *mustn1b*, which mark border zone cardiomyocytes, and again found that cells expressing these genes were mostly in cluster 7 (*Figure 1e* and *Figure 1—figure supplement 4*). Together, these results indicate two things: first, the border zone cardiomyocytes (grouped in cluster 7) can be identified as a separate group in the single-cell RNA-seq data. Secondly, these border zone cardiomyocytes are transcriptionally distinct from remote cardiomyocytes (grouped in cluster 2), while two intermediate cardiomyocyte clusters lie in between.

## Border zone cardiomyocytes resemble embryonic cardiomyocytes

Cardiomyocytes in the border zone disassemble sarcomeric structures and re-express markers of embryonic cardiomyocytes suggesting their dedifferentiation. We therefore wanted to address the level of dedifferentiation of cluster seven cardiomyocytes by comparing their transcriptome with embryonic cardiomyocytes. To obtain embryonic cardiomyocytes we performed FACS sorting on embryos expressing the cardiomyocyte specific marker *Tg(myl7:GFP)*. Single-cell mRNA-sequencing was performed and combined with the single-cell data from the injured adult hearts (*Figure 2a*). The RaceID algorithm identified several cell clusters with separate clusters for the embryonic and adult cardiomyocytes (*Figure 2b, c and d*). Importantly, the cluster seven cardiomyocytes identified in the adult data analysis had a transcriptome that was highly similar to embryonic cardiomyocytes, as shown by pairwise correlation of the differentially expressed genes between the cardiomyocyte clusters: only 257 genes (p-value<0.01), out of 23,786 total detected genes, were differentially expressed between the embryonic and the cluster 7 (border zone) adult cardiomyocytes (*Figure 2d* and *Figure 2—source datas 1*, *2*, *3* and *4*) suggesting a dedifferentiation of border zone cardiomyocytes to embryo–like cells. In contrast, over 1000 genes (p-value<0.01) were differentially expressed between embryonic and cluster 2 (remote zone) adult cardiomyocytes. A heatmap with unbiased hierarchical clustering on the 500 most differentially expressed genes between the three clusters confirmed that cluster 7 cardiomyocytes were more closely related to embryonic than cluster 2 cardiomyocytes (*Figure 2e*). Corroborating the observation that border zone cardiomyocytes resemble embryonic cardiomyocytes we found that genes encoding sarcomere proteins and cardiac-specific factors highly expressed in the embryo were re-expressed in cluster 7 (border zone) cardiomyocytes (*Figure 2f, g and h*).

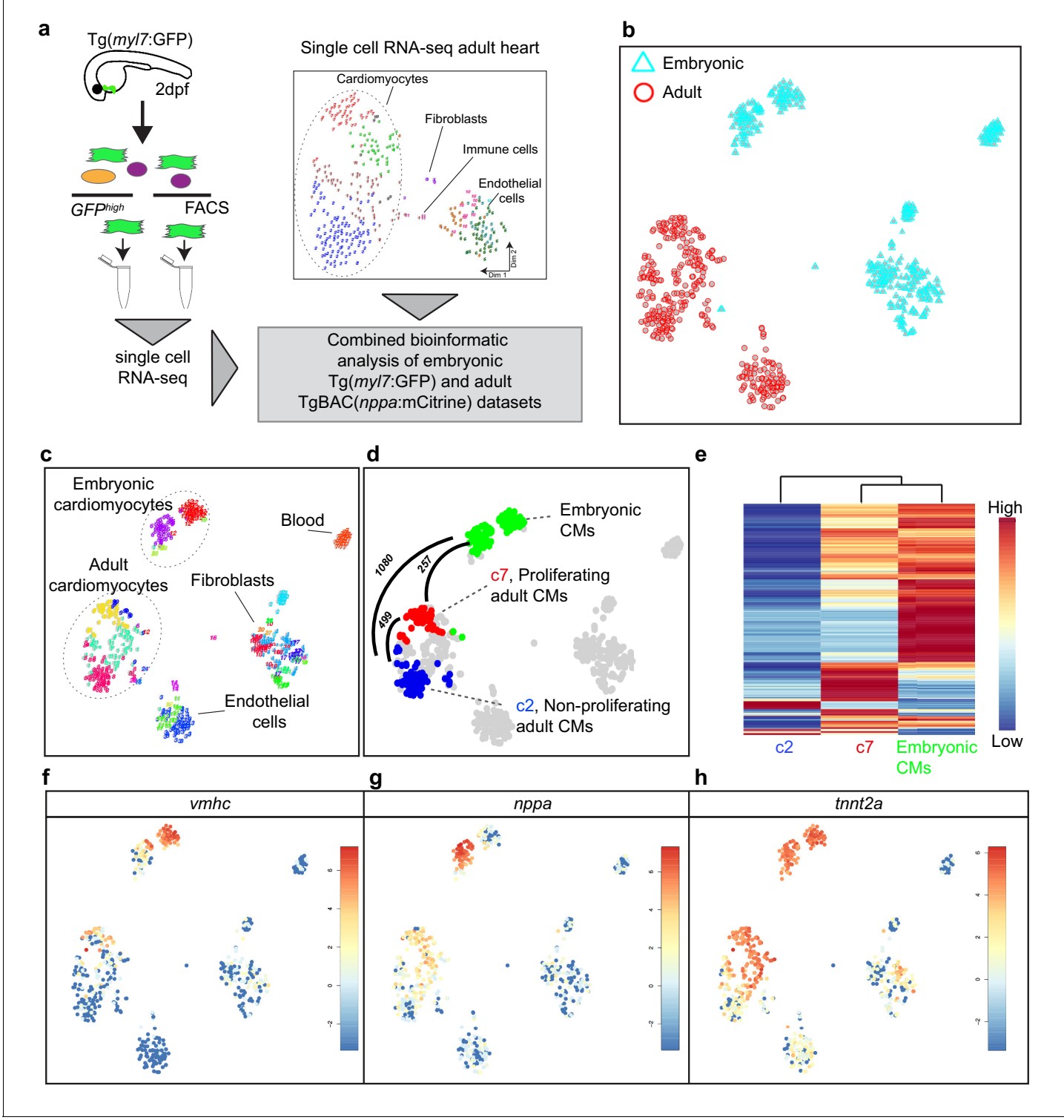

**Figure 2.** Single-cell transcriptome of border zone cardiomyocytes resembles that of embryonic cardiomyocytes. (**a**) Cartoon to illustrate the experimental procedure for single cell analysis of embryonic and adult cardiac cells. (**b**) tSNE map of combined adult (red) and embryonic datasets (light blue). (**c**) tSNE map indicating the different cell types based on marker gene expression. (**d**) tSNE map with the adult cardiomyocytes of the injured heart (cluster 7 in red, cluster 2 in blue and clusters 1 and 4 in gray) and embryonic (2 dpf) cardiomyocytes (in green), with number of pairwise differentially expressed genes (p-value<0.01) indicated between cardiomyocyte clusters. (**e**) Heatmap with hierarchical clustering based on the 500 most differentially expressed genes between clusters. Red color represents high expression, blue color represents low expression. Rows represent individual genes. (**f–h**) tSNE maps visualizing log2-transformed read-counts of *vmhc* (**f**), *nppa* (**g**) and *tnnt2a* (**h**).
*Figure 2 continued on next page*

*Figure 2 continued*

The online version of this article includes the following source data for figure 2:

**Source data 1.** Single-cell mRNA sequencing data from *Tg(cmlc2:GFP)* zebrafish hearts at 2 days post fertilization.
**Source data 2.** List of the clusters from the embryonic heart data (SD5) identified by StemID and the associated cells for each cluster.
**Source data 3.** Combined single-cell mRNA sequencing data from embryonic and adult hearts.
**Source data 4.** List of pairwise differentially expressed genes between all cardiomyocyte clusters from the combined embryonic and adult datasets.

To identify cellular events that occur during this dedifferentiation we used part of the RaceID algorithm (StemID) that uses the single cell transcriptome data and cell clustering to derive a branched lineage tree (*Grün et al., 2016*). The algorithm is based on the premise that stem cells and less differentiated cells tend to exhibit more uniform transcriptomes than differentiated cells, which express smaller numbers of genes at higher rates (*Banerji et al., 2013*). Using this approach, we found large differences in transcriptome entropy, resulting in low (cluster 2), intermediate (clusters 1 and 4) and high (cluster 7) StemID scores (*Figure 3a*). This gradual increase suggests a dedifferentiation axis from cells in cluster 2 (remote myocardium) to cells in cluster 7 (border zone myocardium) and is in good agreement with our finding that the transcriptome of cluster 7 cardiomyocytes resembles an embryonic cardiomyocyte transcriptome (*Figure 3b*). Together, these results indicate that clusters 4 and 7 are enriched for dedifferentiated and proliferative border zone cardiomyocytes while clusters 1 and 2 are enriched for differentiated remote cardiomyocytes.

## Border zone cardiomyocytes induce glycolysis, which is required for proliferation

While cardiomyocytes undergo a well-defined sequence of morphological and transcriptional changes during differentiation, very little is known about the reverse process. Ordering whole-transcriptome profiles of single cells with an unsupervised algorithm can resolve the temporal resolution during differentiation by identifying intermediate stages of differentiation without a priori knowledge of marker genes (*Trapnell et al., 2014*). In this manner, the single-cell mRNA-seq experiment will constitute an in-silico time series, with each cell representing a distinct state of differentiation along a continuum. To analyze the transcriptional changes occurring during this apparent dedifferentiation, the most likely dedifferentiation path, based on the StemID scores, was chosen starting at cluster two and progressing through clusters 1, 4 and 7. Next, gene expression profiles along this pseudo-temporal order were computed for all detected genes using the single-cell transcriptomes. These gene expression profiles were grouped into modules of co-expressed genes using self-organizing maps (SOMs), resulting in 14 modules (*Figure 3c*, *Figure 3—source data 1*). Corroborating our hypothesis of varying differentiation states, we observed that gene expression within these modules changed smoothly over pseudo time. We next analyzed the temporally-ordered expression profiles and identified four groups of genes that shared the same dynamics of expression during this differentiation trajectory. The first group (modules 1, 2) contained genes that were most highly expressed only in cells at the very beginning of the pseudo time line and their expression rapidly declined in cells that were positioned later. This group contained many genes transcribed from mitochondrial DNA and with a role in energy metabolism, which indicates that the cells at the start of the pseudo time line are mature cardiomyocytes. The second group (module 6) contained genes that are induced early and expression stayed constant in cells further along the pseudo time line. Many genes involved in translation and cell cycle regulation follow this expression pattern. The third group showed an exponential increase in expression with the highest expression at the end of the pseudo time line (module 10). This group contained genes with a function in cardiac muscle fiber development and heart contraction. The fourth group (modules 11 and 14) contained genes with a rapid increase in expression that peaks before the end of the pseudo time, suggestive for an early role during the dedifferentiation process. Interestingly this group contained many genes with a known role in glycolysis. Together, these data suggest that border zone cardiomyocytes undergo profound metabolic changes. This was confirmed by GO-term analysis between cluster 7 and cluster 2 cells (*Figure 4—figure supplement 1a and b*, *Figure 1—source data 3*).

To validate the functional consequences of the observed changes in mitochondrial gene expression, we measured succinate dehydrogenase (SDH) enzyme activity, located in the inner

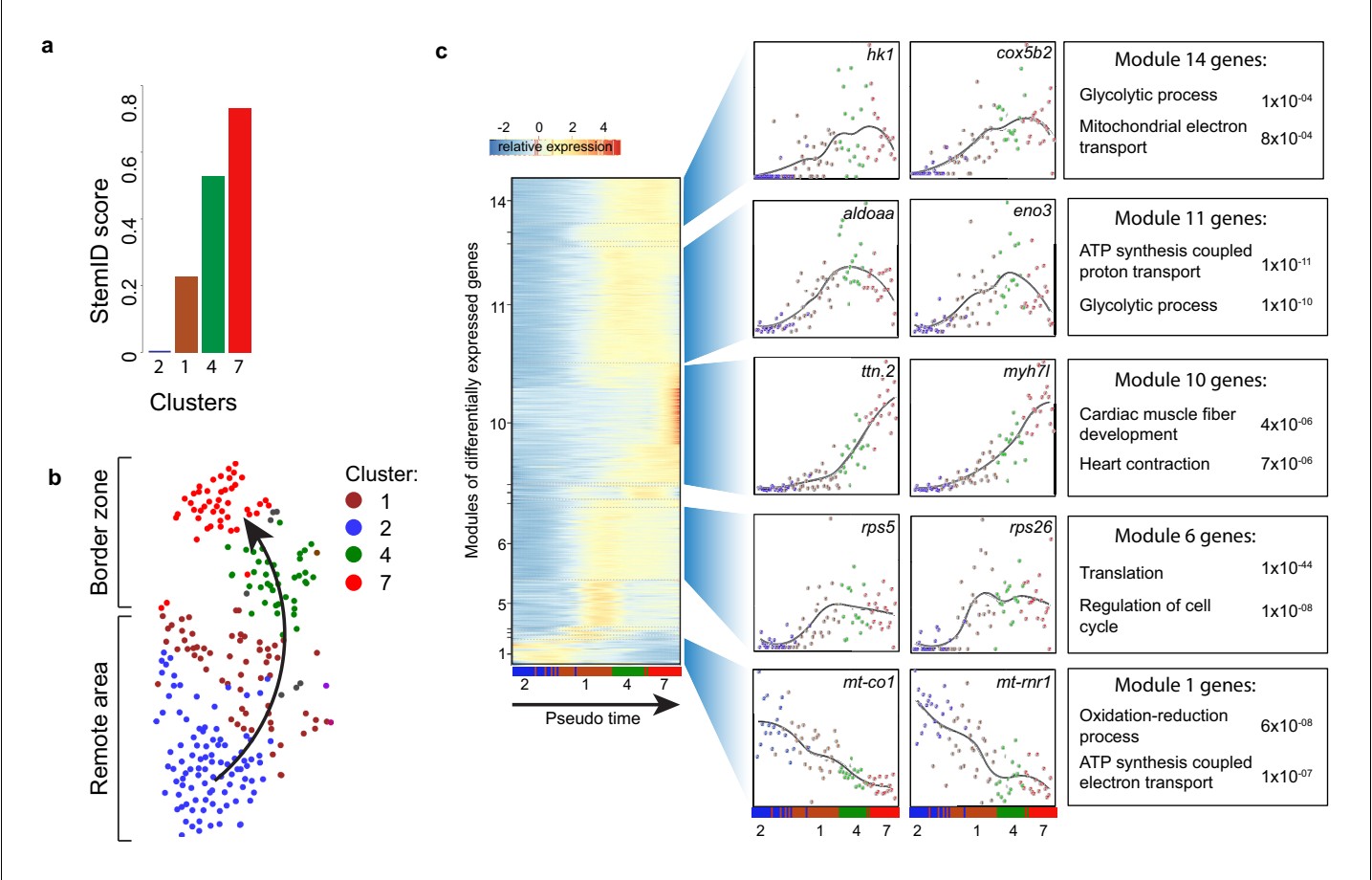

**Figure 3.** Pseudo time analysis reveals dedifferentiation and metabolic changes in border zone cardiomyocytes. (**a**) Bar plot of StemID scores for the cardiomyocyte clusters (clusters #2, 1, 4 and 7) calculated by the formula: number of significant links for each cluster multiplied by the median transcriptome entropy across all cells in a cluster. (**b**) Cardiomyocyte clusters from adult injured heart. Arrow indicates the dedifferentiation path derived from the StemID scores. (**c**) Pseudo time analysis. Left; one-dimensional SOM of z-score transformed expression profiles along the differentiation trajectory incurred by StemID analysis. Y-axis represents the fourteen modules with differentially expressed genes. X-axis represents the pseudo time in which the cells were ordered. Middle; expression profiles of representative genes of the major modules. Y-axis shows transcript counts. X-axis represents the pseudo time. Right; Major gene ontology terms derived from all genes expressed in the module with p-values.
The online version of this article includes the following source data and figure supplement(s) for figure 3:

**Source data 1.** List of genes that are differentially upregulated in the respective modules identified in the pseudo timeline analysis (related to *Figure 3c*).
**Figure supplement 1.** Induction of *hexokinase1* expression precedes expression of *myomesin1b*.

mitochondrial membrane that functions in both the citric acid cycle and electron transport chain. We observed a 40% reduction in SDH activity specifically in the border zone cardiomyocytes as compared to the remote cardiomyocytes (*Figure 4a*). In agreement with the reduced mitochondrial OXPHOS activity, transmission electron microscopy (TEM) imaging revealed more immature mitochondria in border zone cardiomyocytes evidenced by their altered morphology and reduced cristae density (*Figure 4b*), which is consistent with previous reports linking mitochondrial function with morphology (*Giraud et al., 2002*; *Paumard et al., 2002*). Since the pseudotime analysis suggested an upregulation of glycolytic gene expression in the border zone cluster (#7), we performed gene set enrichment analysis (GSEA) for glycolysis genes. The GSEA revealed a strong and significant enrichment in the expression of glycolytic genes in cluster 7 cells compared to cluster 2 cells (*Figure 4—figure supplement 1c*). By in situ hybridization we confirmed the induced expression in border zone cardiomyocytes of the rate-limiting enzymes hexokinase (*hk1*), pyruvate kinase M1/M2a (*pkma*) and pyruvate dehydrogenase kinase (*pdk2a*), which diverts pyruvate away from the TCA cycle (*Figure 4c* and *Figure 1—figure supplement 4b*). Corroborating the suggested enhanced

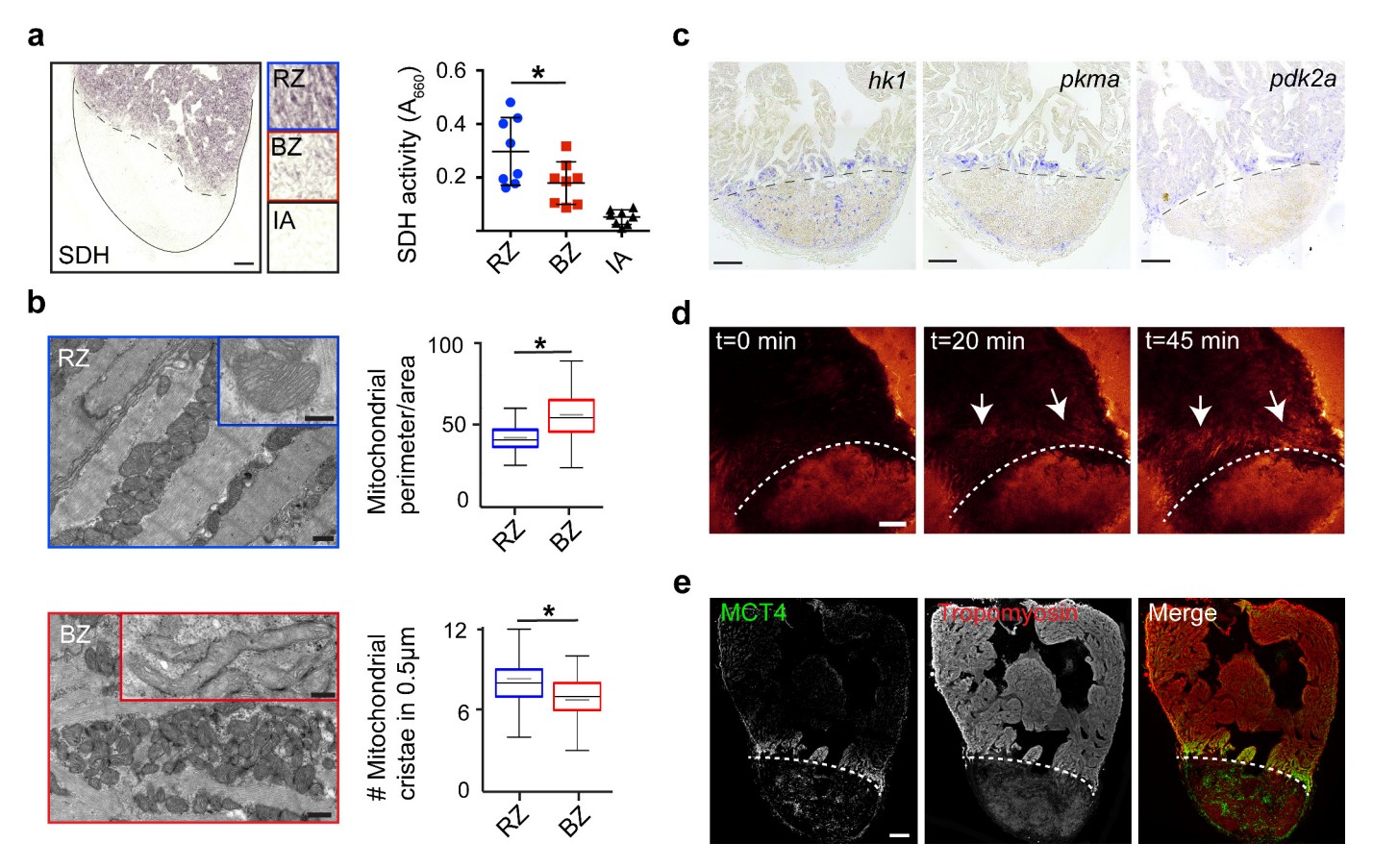

**Figure 4.** Border zone cardiomyocytes undergo a metabolic switch from mitochondrial OXPHOS to glycolysis. (a) Succinate dehydrogenase (SDH) enzyme staining on a seven dpi heart section with injury area separated by dashed line. Quantification of SDH activity in remote zone (RZ), border zone (BZ) and injury area (IA). Scale bar indicates 100 µm. Error bars indicate mean and standard deviation. (b) Transmission electron microscopy (TEM) images of mitochondria in cardiomyocytes from the remote zone and the border zone of a 7 dpi injured heart. Note the disorganized and irregular shaped mitochondria in the border zone cardiomyocyte. Scale bar 500 nm (200 nm in inserts). Graphs show quantification of mitochondrial perimeter-to-area as a measurement for roundness and quantification of mitochondrial cristae density. * p-value<0.05. (c) In situ hybridizations for glycolytic genes *hk1*, *pkma* and *pdk2a* on sections of injured zebrafish hearts at 7 dpi. Dashed line indicates injury site. Scale bars indicate 100 µm. (d) Time-lapse multi-photon confocal images of whole heart. The heart was isolated at 7 dpi and incubated with 2-NBDG, a fluorescent glucose analogue, at t = 0. Dotted line indicates injury area. Arrows point to regions of the border zone. Scale bar represents 100 µm. (e) Confocal image of injured zebrafish hearts at 7 dpi stained for the lactate transporter MCT4 (green) and Tropomyosin (red). Dashed line indicates injury site.

The online version of this article includes the following figure supplement(s) for figure 4:

**Figure supplement 1.** Energy metabolism genes are differentially expressed between cluster 7 and cluster 2 cells.

glycolysis, we observed induced expression of glucose importer genes (*glut1a/slc2a1a* and *glut1b/ slc2a1b*) in cluster 7 cells (*Figure 4—figure supplement 1d*) and enhanced in vivo glucose uptake of border zone cardiomyocytes (*Figure 4d*). Furthermore, genes encoding lactate transporters and their proteins were upregulated in cluster seven and border zone cardiomyocytes (*Figure 4e* and *Figure 4—figure supplement 1d*). Together, these data indicate that during regeneration border zone cardiomyocytes switch energy metabolism from mitochondrial OXPHOS to glycolysis and lactate fermentation.

The pseudo time line analysis suggested that glycolysis genes induction precedes the induction of embryonic cardiac gene expression. Indeed, ISH analysis showed that glycolysis gene expression is already induced at 3 dpi and thereby precedes expression of embryonic cardiac gene expression and cardiomyocyte proliferation, which peaks at 7 dpi (*Figure 3—figure supplement 1*). To address the functional importance of glycolysis we inhibited glycolysis in injured fish with the glucose

analogue 2-Deoxyglucose (2-DG), a general inhibitor of glycolysis, and analyzed its effect on cardio-myocyte proliferation (*Figure 5a*). We observed that repeated injections of 2-DG in the adult zebra-fish with a cryoinjured heart significantly impaired cardiomyocyte proliferation in the border zone (*Figure 5b,c*), suggesting that glycolysis is necessary for cell cycle reentry.

## Nrg1/ErbB2 signaling induces glycolytic gene expression in border zone cardiomyocytes

Next, we investigated the upstream signals that drive the observed metabolic reprogramming in border zone cardiomyocytes during cell cycle reentry. Hypoxia is a well-known stimulus for metabolic reprogramming during cancer and promotes cardiomyocyte proliferation during cardiac

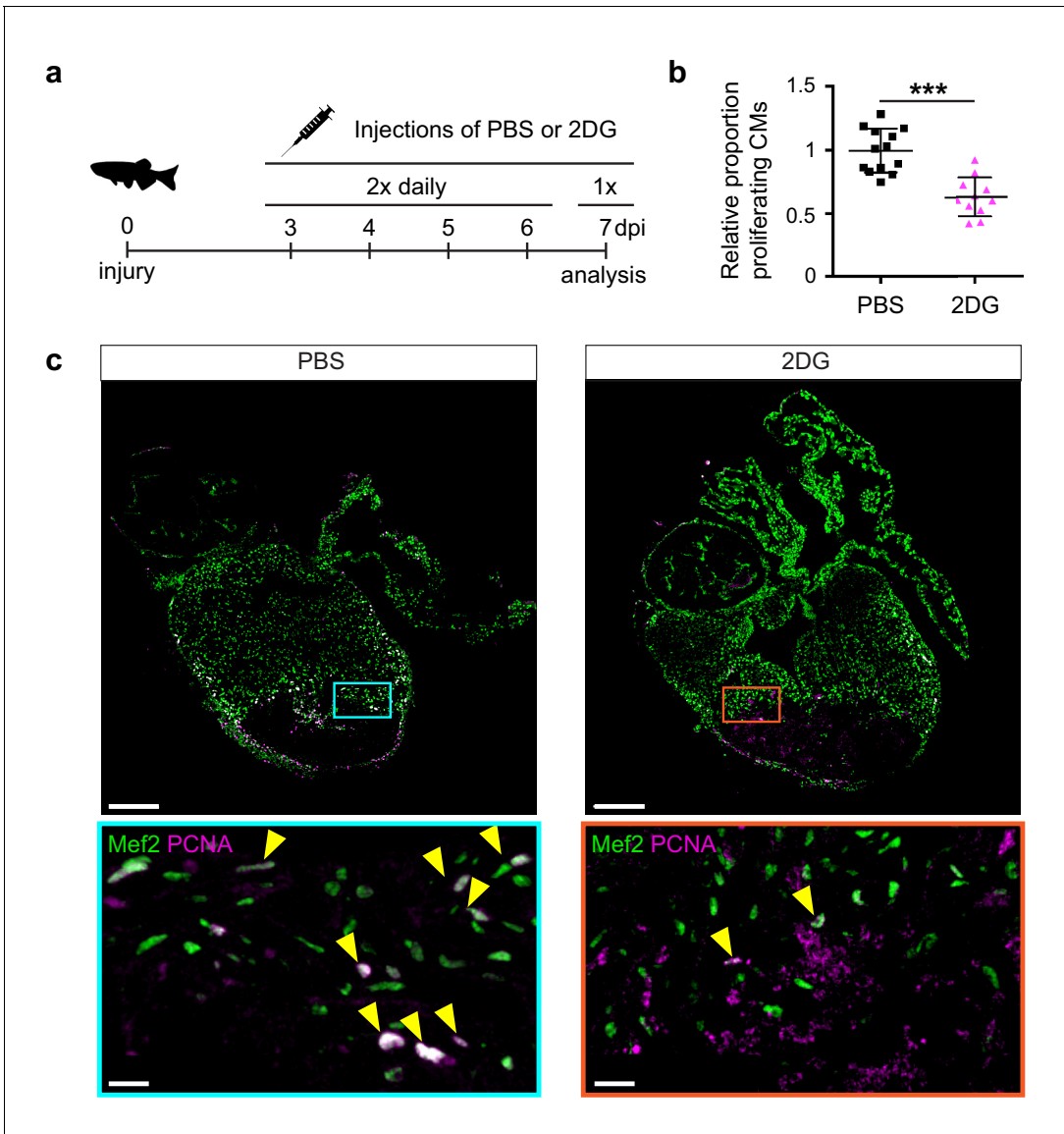

**Figure 5.** 2-Deoxy glucose impairs cardiomyocyte proliferation. (**a**) Experimental design for the 2-DG injections to inhibit glycolysis in injured zebrafish hearts. (**b**) Confocal image of injured zebrafish hearts at seven dpi either injected with PBS or 2-DG stained for Mef2c (green) and PCNA (magenta). Zoom-in images of the borderzone are shown below overview pictures for both PBS (cyan box) and 2-DG (orange box). Arrowheads indicate nuclei positive for Mef2c and PCNA. Scale bar indicates 200 μm (overview) or 20 μm (zoom-in). (**c**) Quantification of the proliferating cardiomyocytes (double Mef2c/PCNA positive) in the border zone of PBS and 2-DG treated hearts represented as the proportion of proliferating cardiomyocytes compared to the average percentage in the PBS injected group. Each dot represents a single heart (three sections per heart analyzed). Hearts were pooled from two separate experiments. Error bars represent mean ± standard deviation. ***, p<0,0001.

regeneration (*Jopling et al., 2012*; *Nakada et al., 2017*; *Vander Heiden et al., 2009*). The responses to hypoxia are induced by the transcription factor, hypoxia inducible factor (HIF), whose activity can be visualized using the *Tg(phd3:GFP)* reporter line (*Santhakumar et al., 2012*; *Wang and Semenza, 1993*). When analysing cryo-injured hearts of *Tg(phd3:GFP)* fish 7 days post injury we did not observe a good correlation between pdh3:GFP reporter activity and the induction of *ldha* expression in border zone cardiomyocytes suggesting that HIF signaling is not required for the observed induction of glycolysis gene expression in border zone cardiomyocytes (*Figure 6—figure supplement 1*). Injury-induced Neuregulin 1 (Nrg1) expression is another potent mitogen that induces cardiomyocyte dedifferentiation and cell cycle reentry by activating ErbB2 receptor signaling (*Gemberling et al., 2015*). Furthermore, in vitro experiments suggest that Nrg1 can induce glucose metabolism (*Cote et al., 2005*; *Suárez et al., 2001*). To address whether Nrg1 can induce metabolic reprogramming in vivo, we used a previously described transgenic zebrafish model in which Nrg1 overexpression (OE) can be induced in a heart specific manner, *tg(cmlc2:CreER; β-act2:BSNrg1)* (*Gemberling et al., 2015*). In this model especially cortical cardiomyocytes start to divide after tamoxifen injection leading to thickening of this layer. We observed a profound and consistent upregulation of glycolysis genes in the cortical myocardium coinciding with the reported cardiomyocyte dedifferentiation and proliferation in this layer (*Figure 6a*). Expression of *ldha* and *hk1*, encoding the rate limiting glycolytic enzyme hexokinase, was strongly induced in the ventricular wall of Nrg1 OE hearts, which correlates well with the observed induction of cardiomyocyte dedifferentiation and proliferation in this region (*Gemberling et al., 2015*). Next, we assessed whether blocking Nrg1/ ErbB2 signaling impairs glycolytic upregulation in the zebrafish border zone after cryoinjury. qPCR confirmed the profound upregulation of glycolytic genes in border zone cardiomyocytes compared to their expression in cardiomyocytes from uninjured hearts (*Figure 6b*). Importantly the ErbB2 inhibitor AG1478 inhibited the induction of glycolytic gene expression in border zone cardiomyocytes (*Figure 6b*). In contrast to the other glycolysis genes, *hk1* expression was not reduced after AG1478 treatment likely as a result of redundant signaling pathways in the border zone. From these results, we conclude that glycolysis gene expression can be induced by Nrg1/ErbB2 signaling even in the absence of cardiac injury and that endogenous Nrg1/ErbB2 signaling is an important mediator of metabolic rewiring during zebrafish heart regeneration.

## Activating ErbB2 signaling induces a metabolic switch from OXPHOS to glycolysis and lactate fermentation in murine cardiomyocytes

The regenerative capacity of the adult murine heart is very low, but cardiomyocyte dedifferentiation and proliferation can be stimulated by cardiomyocyte specific overexpression of a constitutively active ErbB2 receptor (caErbB2 OE) (*D'Uva et al., 2015*). To address whether this is correlated with metabolic changes we performed qPCRs for metabolic genes on cardiac tissue from caErbB2 mice. Indeed, we observed that critical glycolysis genes (e.g. *Pfkp, Pdk3 and Pkm2*) including glucose and lactate transporters (*Slc16A3* and *Slc2A1*) were significantly upregulated in caErbB2 OE cardiomyocytes while genes transcribed from mitochondrial DNA were downregulated (*Figure 6c*). *Pdk3* encodes a pyruvate dehydrogenase kinase, which phosphorylates pyruvate dehydrogenase (PDH). PDH is a mitochondrial multi-enzyme complex that converts pyruvate to Acetyl-CoA and provides a primary link between glycolysis and the TCA cycle. Upon phosphorylation by PDK, p-PDH is inactivated and pyruvate is diverted away from the TCA cycle resulting in enhanced lactate production. Consistent with the increase in *pdk3* expression, an increased phosphorylation of PDH was observed in the inner myocardial layer of caErbB2 OE hearts (*Figure 6d*).

Next, we addressed whether the observed switch in metabolic gene expression correlated with enhanced regenerative capacity after injury. Therefore, we performed myocardial infarction (MI) in wild type and caErbB2 OE hearts. Even though glucose uptake and glycolytic enzyme activity in the ischemic area are increased by MI (*Schelbert and Buxton, 1988*) (*Owen et al., 1969*), we observed a stronger upregulation of glycolytic gene expression and decreased mitochondrial gene expression in caErbB2 OE hearts with MI compared to wild type hearts with MI (*Figure 7—figure supplement 1*). This stronger and consistent upregulation of glycolytic gene expression in caErbB2 OE hearts correlates with the reported enhanced cardiomyocyte proliferation and improved regeneration (*D'Uva et al., 2015*). These findings imply that the enhanced glucose uptake and anaerobic glycolysis observed after MI might be sufficient for cell survival, but that a stronger induction of glycolysis gene expression is required to fully stimulate cardiomyocyte proliferation and regeneration.

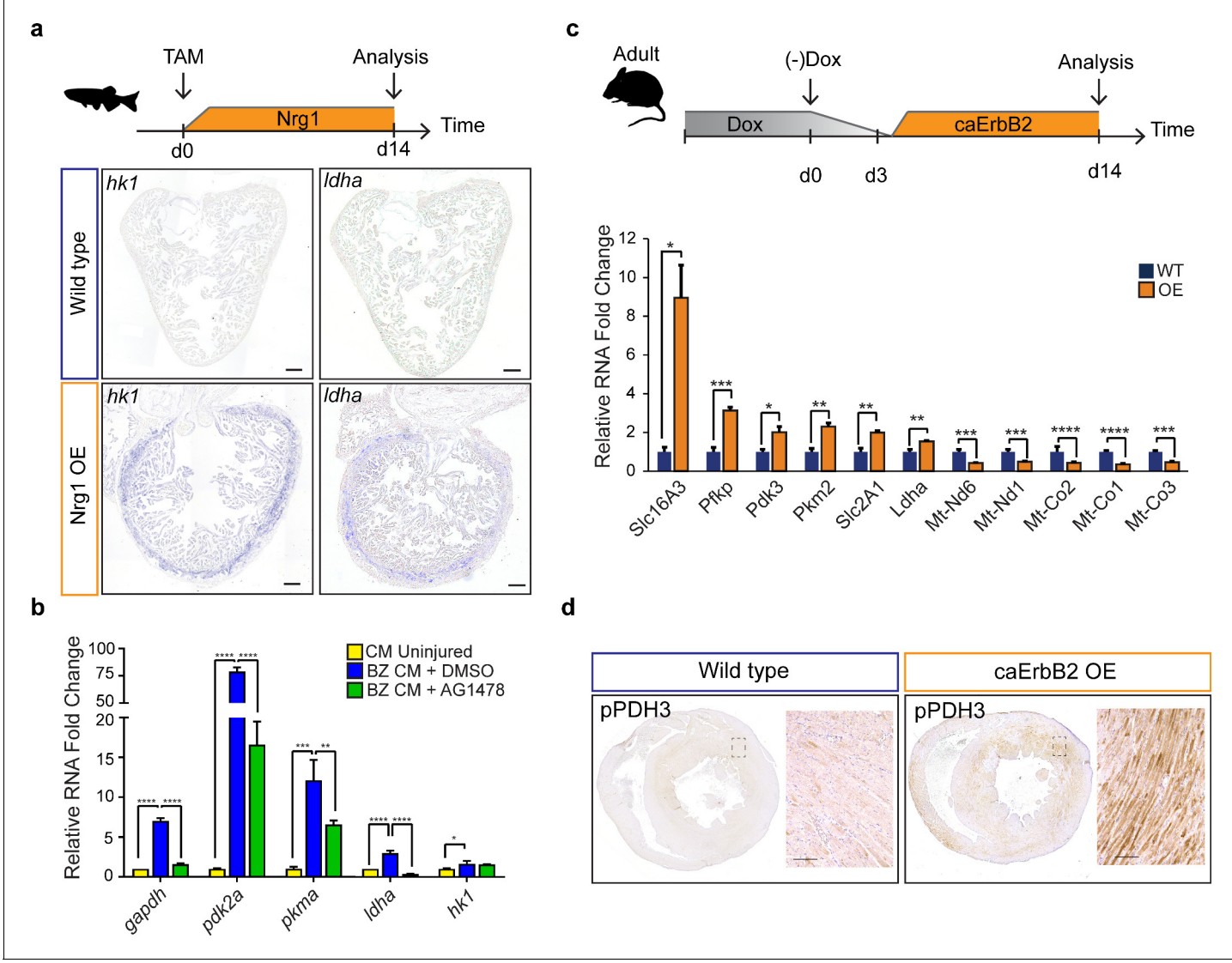

**Figure 6.** Nrg1/ErbB2 signaling induces glycolysis genes in zebrafish and mouse. (**a**) Cartoon showing experimental procedure to induce cardiomyocyte specific Nrg1 expression in zebrafish. Panels show in situ hybridization for *hexokinase 1* (*hk1*) and *lactate dehydrogenase a* (*ldha*) expression on sections of control hearts (*β-act2:BSNrg1*) and Nrg1 OE hearts (*cmlc2:CreER; β-act2:BSNrg1*). Scale bars represent 100 μm. (**b**) qPCR results for glycolytic genes showing their relative fold change in DMSO treated (n = 9) (blue) and AG1478 treated (n = 9) (green) nppa:mCitrine high border zone cardiomyocytes at 3dpi compared to uninjured adult cardiomyocytes (n = 4) (yellow). Error bars represent standard deviation. (**c**) Upper panel: Cartoon showing the experimental procedure to analyse metabolic gene expression after activating ErbB2 signaling in the murine heart. Lower panel: qPCR results for metabolic genes showing their relative fold change in caErbB2 OE (n = 4) heart compared to control WT hearts (n = 4). Error bars represent standard deviation. (**d**) Immunohistochemistry for phospho-PDH3 on sections of control and caErbB2 OE hearts. Scale bars represent 100 μm. *=p < 0.05, **=p < 0.01, ***=p < 0.001, ****=p < 0.0001.

The online version of this article includes the following source data and figure supplement(s) for figure 6:

**Source data 1.** Oligonucleotide sequences for real-time PCR analysis.

**Figure supplement 1.** Hypoxia sensor *phd3*:GFP expression does not correlate with expression of the glycolytic gene *ldha*.

## CaErbB2 induced cardiomyocyte proliferation depends on glycolysis

Finally, we addressed whether the observed metabolic switch to glycolysis in caErbB2 OE cardiomyocytes is required for their reentry into the cell cycle (*Figure 7a*). Corroborating our model, we indeed observed that treating caErbB2 OE cardiomyocytes in vitro with the glycolysis inhibitors 2-DG or lonidamine strongly and significantly impaired cell cycle reentry (*Figure 7b and c*) and cytokinesis (*Figure 7d and e*). Together these results indicate that in murine cardiomyocytes, ErbB2

signaling drives a metabolic switch towards glycolysis, which is required for their cell cycle reentry. These results suggest that this metabolic switch in cardiomyocytes is beneficial for heart regeneration.

## Discussion

Heart regeneration in zebrafish is very efficient and relies on the proliferation of preexisting cardiomyocytes. The number of proliferating cardiomyocytes that is induced by injury is very small and

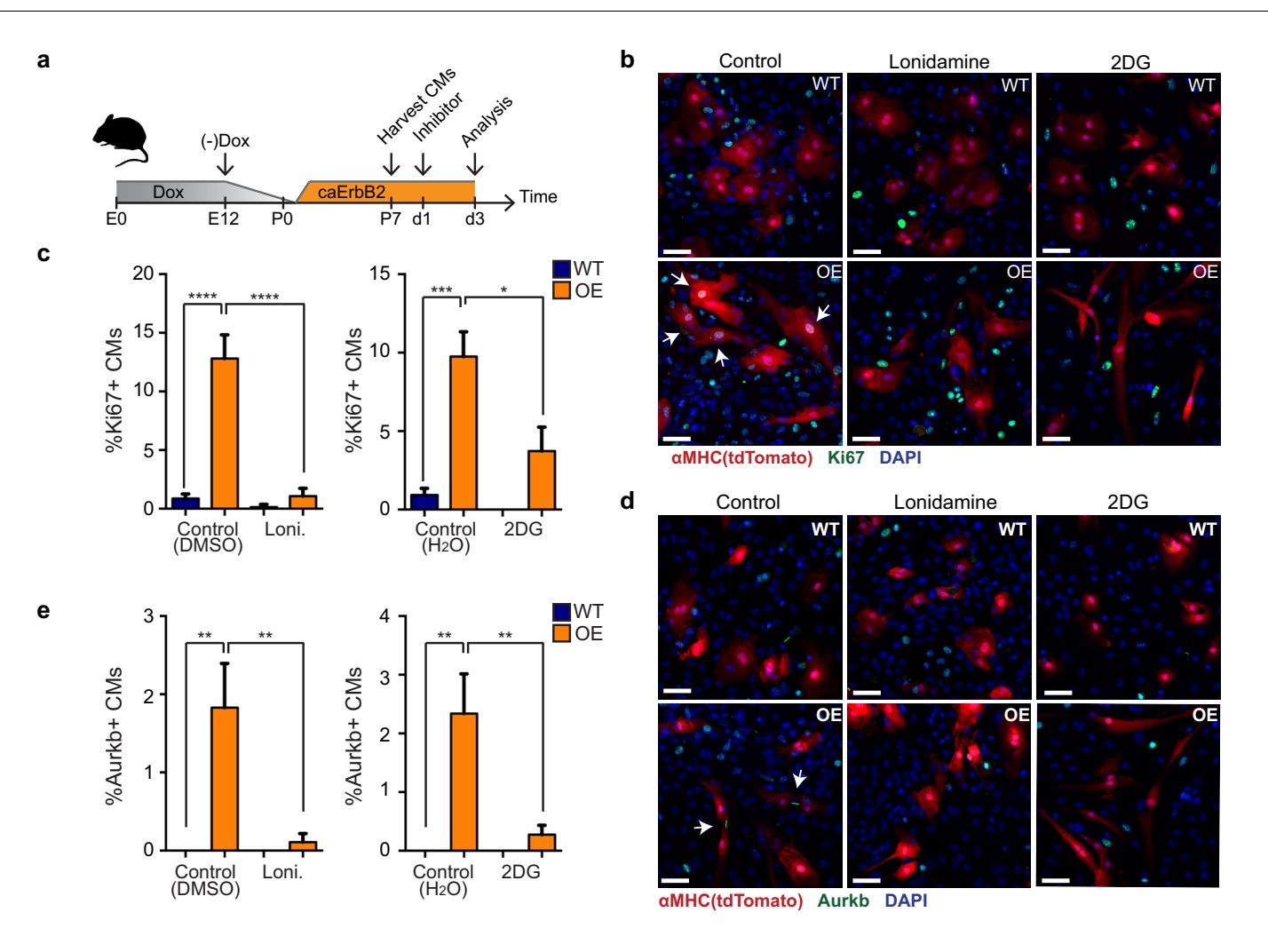

**Figure 7.** Glycolysis inhibitors impair mitogenic effect of ErbB2 activation in cardiomyocytes. (a) Cartoon showing the experimental procedure to analyse the effects of glycolysis inhibitors (2-DG and lonidamine) on cardiomyocyte proliferation. (b) Immunofluorescence analysis on P7 cardiac cultures derived from WT and caErbB2 OE hearts that are endogenously fluorescent for tdTomato under the *MYH6* promoter, stained for the cell-cycle marker Ki67. Arrows point at Ki67+ CMs. (c) Quantification of % Ki67+ CMs from WT and caErbB2 OE derived from P7 cardiac cultures treated with the glycolysis inhibitors 2-DG (n = 4 for WT and n = 4 for OE), and lonidamine (n = 7 for WT and n = 4 for OE) or their diluents as controls. (d) Immunofluorescence analysis on P7 cardiac cultures derived from WT and caErbB2 OE hearts that are endogenously fluorescent for tdTomato under the *MYH6* promoter, stained for the cytokinesis marker Aurora kinase B. Arrows point at Aurkb+ CMs. (e) Quantification of % Aurkb+ CMs from WT and caErbB2 OE derived from P7 cardiac cultures treated with the glycolysis inhibitors 2-DG (n = 4 for WT and n = 4 for OE), and lonidamine (n = 7 for WT and n = 4 for OE) or their diluents as controls. In all panels, bars represent the mean, and error bars represent s.e.m *p<0.05, **p<0.01, ***p<0.001, ****p<0.0001, Scale bars represent 50 µm.

The online version of this article includes the following figure supplement(s) for figure 7:

**Figure supplement 1.** Activation of ErbB2 signaling in murine MI model induces glycolytic gene expression while repressing mitochondrial genes.

they comprise only a small portion of the total cardiomyocytes, which has hampered their characterization. Thus far, RNA-seq data sets and differential gene analysis was based on whole tissue preparations or cryosections of uninjured and injured hearts (*Goldman et al., 2017 Kang et al., 2016*; *Lai et al., 2017*; *Lien et al., 2006*; *Sleep et al., 2010*; *Wu et al., 2016*). Not only cardiomyocytes but other cell types in the heart such as epicardial and endocardial cells respond to the injury by the upregulation of injury-induced genes (*Lepilina et al., 2006*). Furthermore, the injured heart is infiltrated by immune cells and fibroblasts appear (*González-Rosa et al., 2011*; *Lai et al., 2017*). Together, this complicates the detection of cardiomyocyte specific gene responses. The use of single cell transcriptomics can overcome these limitations and has allowed us to identify and characterize the different cardiomyocyte populations in the regenerating zebrafish heart. Here we have generated a unique RNA-seq dataset at single-cell resolution of cardiomyocytes during heart regeneration, which can be used as a resource to study and identify novel mechanisms.

Activation of Yap signaling in the murine heart promotes cardiomyocyte proliferation, which involves a partial reprogramming and dedifferentiation of adult cardiomyocytes towards a neonatal fate (*Monroe et al., 2019*). Interestingly, single-cell RNA-seq analysis of connective tissue cells during limb blastema formation revealed that their transcriptome is highly similar to embryonic limb precursor cells (*Gerber et al., 2018*). Together with our results that a similar reprogramming takes place during injury induced heart regeneration in zebrafish it suggests a common theme in which adult tissues that lack a clear stem cell use an alternative strategy to regenerate the missing tissue by reprogramming differentiated cells into embryo-like cells. This might involve different mechanisms since Yap signaling is not required for cardiomyocyte proliferation during zebrafish heart regeneration (*Flinn et al., 2019*). The presence of embryo-like cardiomyocytes in the border zone after injury can explain the observed switch from trabecular cardiomyocytes into cortical cardiomyocytes during zebrafish heart regeneration, since embryonic cardiomyocytes give rise to both trabecular and cortical cardiomyocytes (*Sánchez-Iranzo et al., 2018*; *Staudt et al., 2014*).

Increased mitochondrial OXPHOS activity promotes cardiomyocyte maturation and reduces the proliferative capacity of cardiomyocytes (*Mills et al., 2017*). This correlates well with the loss of regenerative capacity of the murine heart in the first week after birth at which time the metabolism in cardiomyocytes changes from predominantly glycolysis to mitochondrial OXPHOS (*Lopaschuk et al., 1992*; *Menendez-Montes et al., 2016*; *Porrello et al., 2011*). Reactive oxygen species (ROS) generated by mitochondrial OXPHOS induce DNA damage causing cardiomyocyte cell-cycle arrest (*Nakada et al., 2017*; *Puente et al., 2014*; *Tao et al., 2016*). Our pseudo time line analysis shows a rapid downregulation of mitochondrial gene expression together with an increase in genes encoding glucose transporters, glycolytic enzymes and lactate transporters. A direct role for glycolysis and lactate fermentation in cardiomyocyte proliferation had not been addressed and our results that pharmacological inhibition of glycolysis impairs cardiomyocyte proliferation suggest that activation of glycolysis can drive the reprogramming of cardiomyocytes. It also raises the question why cardiomyocytes need to switch their metabolism to glycolysis to reenter the cell cycle? Interestingly, a similar metabolic shift from mitochondrial OXPHOS to glycolysis and lactate fermentation occurs in proliferating tumor cells, and was first described by Otto Warburg (*Warburg et al., 1927*). Since this glucose to lactate transformation occurs regardless of whether oxygen is present it is referred to as aerobic glycolysis. While glycolysis generates much less ATP compared to fatty acid oxidation, it is thought that glycolysis and the connected pentose-phosphate pathway provides essential metabolites that are needed to create sufficient biomass to sustain proliferation of the tumor cells (*Vander Heiden et al., 2009*). Furthermore, progenitor cells in the developing embryo as well as induced pluripotent stem cells depend on glycolysis to maintain proliferation and their potency (*Folmes et al., 2011*; *Gu et al., 2016*; *Mathieu and Ruohola-Baker, 2017*). In addition, glycolytic enzymes such as PKM2 and PFKFB4 can also directly interact with cell cycle regulators to promote proliferation (*Yang et al., 2011*; *Dasgupta et al., 2018*). The precise role for glycolysis in driving the cellular reprogramming during heart regeneration needs to be further investigated using genetic loss- and gain-of-function experiments combined with metabolomics, which is challenging given the low number of proliferating cardiomyocytes in the regenerating heart.

Activation of Nrg1/ErbB2 signaling in either zebrafish or mouse hearts induces cardiomyocyte dedifferentiation and proliferation (*D'Uva et al., 2015*; *Gemberling et al., 2015*). Our results build upon these observations and indicate that Nrg1/ErbB2 signaling induces profound metabolic reprogramming in cardiomyocytes and that this is required for efficient cardiomyocyte proliferation.

Future work should address how Nrg1/ErbB2 signaling induces metabolic reprogramming in cardio-myocytes, which could have important implications for stimulating mammalian heart regeneration (*Polizzotti et al., 2015*).

# Materials and methods

## Key resources table

| Reagent type (species) or resource | Designation | Source or reference | Identifiers | Additional information |
|---|---|---|---|---|
| Strain, strain background (*Danio rerio*) | Tupfel Long Fin (TL) | ZIRC | ZDB-GENO-990623–2 | |
| Genetic reagent (*Danio rerio*) | Tg(*myl7*:dsRED)^s879Tg | *Chi et al., 2008* | ZDB-FISH-150901–3078 | |
| Genetic reagent (*Danio rerio*) | Tg(*myl7*:GFP)^twu34Tg | *Huang et al., 2003* | ZDB-FISH-150901–212 | |
| Genetic reagent (*Danio rerio*) | Tg(*phd3*:GFP)^sh144 | *Santhakumar et al., 2012* | ZDB-FISH-150901–26851 | |
| Genetic reagent (*Danio rerio*) | Tg(*gata4*:EGFP)^ae1 | *Heicklen-Klein and Evans, 2004* | ZDB-FISH-150901–14762 | |
| Genetic reagent (*Danio rerio*) | TgBAC (*nppa:mCitrine*) | This paper | | More info on generation of this line can be obtained from the Materials and methods section 'Transgenic zebrafish lines and cryoinjury'. |
| Cell line (*Mus musculus*) | TetRE-caErbb2 X MYH6-tTA X MYH6-cre ROSA26-tdTomato | *D'Uva et al., 2015* | | More info on how to obtain primary cell line can be obtained from Materials and methods sections 'Transgenic mouse lines and animal procedures' and 'Pharmacological inhibition of glycolysis'. |
| Biological sample (*Mus musculus*) | TetRE-caErbb2 X MYH6-tTA hearts | *D'Uva et al., 2015* from: *Xie et al., 1999* and *Yu et al., 1996*. | The Jackson Laboratory, stock no. 010577; | |
| Biological sample (*Danio rerio*) | Tg(cmlc2:CreER)^pd10 x Tg(β-act2:BSNrg1)^pd107 hearts | *Gemberling et al., 2015* | ZDB-FISH-150901–25249 x ZDB-FISH-150901–25354 | |
| Biological sample (*Danio rerio*) | Tg(*gata4*:EGFP)^ae1 hearts | Lin et al. 2009 | ZDB-FISH-150901–14762 | |
| Biological sample (*Danio rerio*) | Tg(*phd3*:GFP)^sh144 hearts | *Santhakumar et al., 2012* | ZDB-FISH-150901–26851 | |
| Antibody | Anti-AuroraB kinase (mouse monoclonal) | BD Transduction Laboratories | #611082, RRID:AB_2227708 | 1:200 |
| Antibody | Anti-Ki67 (Rabbit monoclonal) | Cell Marque | #275R | 1:200 |

*Continued on next page*

*Continued*

| Reagent type (species) or resource | Designation | Source or reference | Identifiers | Additional information |
|---|---|---|---|---|
| Antibody | Anti-MCT4 (rabbit polyclonal) | Santa Cruz | #SC50329, RRID:AB_2189333 | 1:200 |
| Antibody | Anti-PCNA (mouse monoclonal) | DAKO | #M0879, RRID:AB_2160651 | 1:800 |
| Antibody | Anti-GFP (chicken polyclonal) | Aves | #GFP-1010, RRID:AB_2307313 | 1:1000 |
| Antibody | Anti-Mef2C (rabbit polyclonal) | Santa Cruz/Biorbyt | #SC313, RRID: AB_631920 / #orb256682 | Both 1:1000 |
| Chemical compound, drug | 2-Deoxyglucose | Sigma-Aldrich | #D6134 | 1 mg/g or 3 mM |
| Chemical compound, drug | Lonidamine | Sigma-Aldrich | L4900 | 80 uM |
| Chemical compound, drug | 2NBDG | Caymanchem | #11046 | 400 uM |
| Chemical compound, drug | AG1478 | Selleck Chemical | S2728 | 5M, from 10 mM stock in DMSO |
| Chemical compound, drug | Doxycycline | Harlan Laboratories | TD02503 | |
| Chemical compound, drug | Trizol | Life technologies | #15596026 | |
| Chemical compound, drug | Fast SYBR Green Master Mix | Applied Biosystems | #4385612 | |
| Chemical compound, drug | Collagenase type II | Gibco | 17101015 | 0.1% |
| Chemical compound, drug | TrypLE Express Enzyme (1x), phenol red | Gibco | 12605036 | |
| Software, algorithm | FIJI | *Schindelin et al., 2012* | Version 2.0.0, RRID:SCR_002285 | |
| Software, algorithm | RaceID2/StemID | Grün, D. et al. De Novo Prediction of Stem Cell Identity using Single-Cell Transcriptome Data. Cell Stem Cell 19, 266–277 (2016). | RRID:SCR_017242 | |
| Software, algorithm | Rstudio | Rstudio | RRID: SCR_000432 | Version 1.2.1335 |
| Software, algorithm | Imaris | Bitplane | RRID: SCR_0007370 | V9.3.1 |
| Software, algorithm | Gene Set Enrichment Analysis | Genepattern, Broad Institute | RRID: SCR_003199 | # of permutations = 1000 |

*Continued on next page*

*Continued*

| Reagent type (species) or resource | Designation | Source or reference | Identifiers | Additional information |
|---|---|---|---|---|
| Commercial assay, kit | Superscript III First strand synthesis system | Thermo Fisher Scientific | #18080051 | Input 200 ng RNA |
| Commercial assay, kit | MiRNeasy | Qiagen | 217004 | |
| Commercial assay, kit | High capacity cDNA Reverse Transcription kit | Applied Biosystems | 4374966 | Input 1 ug RNA |
| Commercial assay, kit | Neonatal Dissociation kit | Miltenyi Biotec | 130-098-373 | |
| Commercial assay, kit | TruSeq small RNA primers | Illumina | 20005613 | |

## Transgenic zebrafish lines and cryoinjury

The following fish lines were used: TL, Tg(*phd3*:GFP), Tg(*gata4*:EGFP)[ae1] and myl7:GFP[twu34Tg] (*Huang et al., 2003*) (*Kikuchi et al., 2010*; *Santhakumar et al., 2012*). The *Tg(cmlc2:CreER; β-act2: BSNrg1)* line was used as described before (*Gemberling et al., 2015*). The *TgBAC(nppa*:mCitrine) line was generated essentially as described previously (*Bussmann and Schulte-Merker, 2011*). In short, an iTOL2_amp cassette for pTarBAC was inserted in the vector sequence of bacterial artificial chromosome (BAC) CH211-70L17, which contains the full *nppa* locus. Subsequently, a mCitrine_kan cassette was inserted at the ATG start codon of the first exon of the nppa gene. Amplification from a pCS2+mCitrine_kanR plasmid was achieved with primers:

FWD_NPPA_HA1_GFP

5′-gagccaagccagttcagagggcaagaaaacgcattcagagacactcagagACCATGGTGAGCAAGGGCGAGG-3′ and REV_NPPA_HA2_NEO

5′-gtctgctgccaaaccaggagcagcagtcctgtcagaattagtcccccggcTCAGAAGAACTCGTCAAGAAGGCGA TAGAA −3′.

Sequences homologous to the BAC are shown in lower case. Recombineering was performed following the manufacturer's protocol (Red/ET recombination; Gene Bridges GmbH, Heidelberg, Germany) with minor modifications. BAC DNA isolation was carried out using a Midiprep kit (Life Technologies BV, Bleiswijk, The Netherlands). BAC DNA was injected at a concentration of 300 ng/μl in the presence of 25 ng Tol2 mRNA. At three dpf, healthy embryos displaying robust nppa-specific fluorescence in the heart were selected and grown to adulthood. Subsequently, founder fish were identified by outcrossing and their progeny grown to adulthood to establish the transgenic line.

Zebrafish of ~4 to 18 months of age (males and females, TL strain) were used for regeneration experiments. Cryoinjuries were performed as previously described (*Schnabel et al., 2011*), except that a liquid nitrogen-cooled copper filament of 0.3 mm diameter was used instead of dry ice.

Sample sizes were chosen to accommodate the generally accepted standards in the field: five or more cryoinjured hearts per condition.

Animals were only excluded from experiments in case of severe sickness/infection/aberrant behavior (according to animal experiment guidelines).

## Transgenic mouse lines and animal procedures

Doxycycline-inducible CM-restricted overexpression of a constitutively active Erbb2 (caErbb2) was generated by crossing the TetRE–caErbb2 (*Xie et al., 1999*) mouse line with *MYH6*– tTA which expresses the tetracycline-responsive transcriptional activator (tTA) under the control of the human alpha myosin heavy chain promoter (*MYH6*) (*Yu et al., 1996*). Doxycycline (DOX, Harlan Laboratories, TD02503) was administered in the food to repress transgene expression. For cultures derived of OE/WT hearts, we additionally intercrossed the *MYH6*-cre ROSA26-tdTomato transgenes in order to visualize CMs.

For myocardial infarction, mice were sedated with isoflurane (Abbott Laboratories) and were artificially ventilated following tracheal intubation. Experimental myocardial infarction was induced by

ligation of the left anterior descending coronary artery (LAD ligation). Following the closure of the thoracic wall mice were warmed for several minutes until recovery.

## Immunofluorescence

ADULT: For immunofluorescence, hearts were extracted, fixed in 4% PFA at room temperature for 1,5 hr and cryosectioned into 10 µm sections. Heart sections were equally distributed onto seven serial slides so each slide contained sections representing all areas of the ventricle.

Primary antibodies used were anti-AuroraB kinase (BD Transduction laboratories #611082, 1:200), anti-Ki67 (Cell Marque #275R, 1:200), anti-MCT4 (Santa Cruz #SC50329, 1:200), anti-PCNA (Dako #M0879, 1:800), anti-GFP (aves #GFP-1010, 1:1000), and anti-Mef2c (Santa Cruz #SC313, Biorbyt #orb256682 both 1:1000). Antigen retrieval was performed by heating slides containing heart sections at 85°C in 10 mM sodium citrate buffer (pH 6) for 10 min. Secondary antibodies conjugated to Alexa 488 (ThermoFisher Scientific), Cy3 or Cy5 (Jackson Laboratories) were used at a dilution of 1:1000. Nuclei were shown by DAPI (4',6-diamidino-2-phenylindole) staining. Images of immunofluorescence stainings are single optical planes acquired with a Leica Sp8 or Sp5 confocal microscope. Quantifications of PCNA, Mef2, and mCitrine expression were performed in cardiomyocytes situated within 150 µm from the wound border on 3 sections of >4 hearts. Sections were masked before quantification.

EMBRYONIC: Live embryos were immobilized using ms222 and embedded in nitrocellulose + E3 to be mounted on a Leica SPE confocal microscope, followed by a Z-stack maximum projection (step size 2 µm).

MAMMALIAN P7 CARDIAC CULTURES: For immunofluorescence, cardiac cultures were fixed with 4% PFA for 10 min on room temperature on the shaker, followed by permeabilization with 0.5% Triton X-100 in PBS for 5 min, and blocking with 5% bovine serum albumin (BSA) in PBS containing 0.1% Triton for 1 hr at room temperature. Masking was performed before quantification.

## Quantitative PCR

ADULT ZEBRAFISH: *nppa:*mCitrine zebrafish were cryoinjured and received two overnight pulses of DMSO (1:2000) or 5 µM AG1478 (10 mM stock in DMSO; Selleck Chemical, Houston, TX) from 1dpi to 2dpi and from 2dpi to 3dpi. Then, hearts were extracted for both conditions (n = 9) as well as uninjured *cmlc2:*dsRED controls (n = 4), dissociated and single cells were FACS sorted for mCitrine and dsRED expression respectively. RNA was isolated from sorted cells using Trizol (Life Technologies BV, Bleiswijk, The Netherlands). RNA was quantified using a NanoDrop spectrophotometer. Superscript III First Strand Synthesis System (ThermoFisher Scientific) was used to reverse transcribe 200 ng of purified RNA per condition following manufacturer's protocol. qPCR reactions were performed using Fast SYBR Green PCR Master Mix (Applied Biosystems). Oligonucleotide sequences for real-time PCR analysis performed in this study are listed in *Figure 6—source data 1*.

MOUSE: RNA from whole hearts was isolated using the MiRNeasy kit (Qiagen, 217004), according to the manufacturer's instructions. RNA was quantified using a NanoDrop spectrophotometer. A High Capacity cDNA Reverse transcription kit (Applied Biosystems, 4374966) was used to reverse transcribe 1 µg of purified RNA according to the manufacturer's instructions. qPCR reactions were performed using Fast SYBR Green PCR Master Mix (Applied Biosystems). Oligonucleotide sequences for real-time PCR analysis performed in this study are listed in *Figure 6—source data 1*.

## In situ hybridization

PARAFFIN: After o/n fixation in 4% PFA, hearts were washed in PBS twice, dehydrated in EtOH, and embedded in paraffin. Serial sections were made at 10 µm. In situ hybridization was performed on paraffin-sections as previously described (*Moorman et al., 2001*) except that the hybridization buffer used did not contain heparin and yeast total RNA.

CRYOSECTIONS: Sections were obtained as described earlier. In situ hybridization was performed as for paraffin, however sections were pre-fixed for 10 min in 4% PFA + 0.25% glutaraldehyde before Proteinase K treatment. Moreover, slides were fixed for 1 hr in 4% PFA directly after staining. When in situ hybridization was combined with immunofluorescence, Fast Red staining solution was used instead of NBT-BCIP.

## Isolation of single cells from cryoinjured hearts

Cryoinjured hearts (n = 13) were extracted at seven dpi. Cells were dissociated according to *Tessadori et al. (2012)*. For cell sorting, viable cells were gated by negative DAPI staining and positive YFP-fluorescence. In brief, the FACS gating was adjusted to sort cells for nppa:mCitrine[high] (to enrich for proliferating cardiomyocytes) and nppa:mCitrine[low] (remote cardiomyocytes and other cell types) cells. In total n = 576 mCitrine[high] and n = 192 mCitrine[low] cells were sorted into 384-well plates and processed for mRNA sequencing as described below.

## Isolation of single cells from embryonic zebrafish

Transgenic *tg(myl7:GFP)* 2-day-old embryos (n = 200) were dechorionated and digested in HBSS $Ca^{2+}/Mg^{2+}$ free media containing 0.1% collagenase type II (Gibco) at 32°C for 30–40 min followed by 1X TrypLE Express (Gibco) for 15–30 min at 32°C with agitation. Dissociated cells were then FAC-Sorted and subjected to single-cell mRNA-seq.

## Single-cell mRNA sequencing

Single-cell sequencing libraries were prepared using SORT-seq (*Muraro et al., 2016*). Live cells were sorted into 384-well plates with Vapor-Lock oil containing a droplet with barcoded primers, spike-in RNA and dNTPs, followed by heat-induced cell lysis and cDNA syntheses using a robotic liquid handler. Primers consisted of a 24 bp polyT stretch, a 4 bp random molecular barcode (UMI), a cell-specific 8 bp barcode, the 5' Illumina TruSeq small RNA kit adapter and a T7 promoter. After cell-lysis for 5 min at 65 °C, RT and second strand mixes were distributed with the Nanodrop II liquid handling platform (Inovadyne). After pooling all cells in one library, the aqueous phase was separated from the oil phase, followed by IVT transcription. The CEL-Seq2 protocol was used for library prep (*Hashimshony et al., 2016*). Illumina sequencing libraries were prepared with the TruSeq small RNA primers (Illumina) and paired-end sequenced at 75 bp read length on the Illumina NextSeq platform. Mapping was performed against the zebrafish reference assembly version 9 (Zv9).

## Bioinformatic analysis

To analyze the single-cell RNA-seq data, we used the previously published RaceID algorithm (*Grün et al., 2015*). For the adult hearts we had a dataset consisting of two different libraries of 384 cells each for a combined dataset of 768 cells, in which we detected 19257 genes. We detected an average of 10,443 reads per cell. Based on the distribution of the log10 total reads plotted against the frequency, we introduced a cutoff at minimally 3500 reads per cell before further analysis. This reduced the number of cells used in the analysis to 352. Next, we downsampled reads to 3500 unique (UMI corrected) transcripts per cell, as means of normalization. Moreover, we discarded genes that were not detected at >3 transcripts in >1 cell and These cutoffs is a stringent normalization method that allows us to directly compare detected transcripts between cells from different cell types and libraries. Batch-effects were analyzed and showed no plate-specific clustering of certain clusters.

For embryonic cardiomyocytes, two libraries of 384 cells were combined to obtain a set of 768 cells, in which 22271 genes could be detected. An average of 4412 reads per cell was detected. After downsampling of this library to 3500 reads per cell 302 cells were included for further analysis. Further analysis was performed by combining the embryonic heart data with the injured adult heart data.

The StemID algorithm were used as previously published (*Grün et al., 2016*). In short, StemID is an approach developed for inferring the existence of stem cell populations from single-cell transcriptomics data. StemID calculates all pairwise cell-to-cell distances (1 – Pearson correlation) and uses this to cluster similar cells into clusters that correspond to the cell types present in the tissue. The StemID algorithm calculates the number of links between clusters. This is based on the assumption that cell types with less links are more canalized while cell types with a higher number of links have a higher diversity of cell fates. Besides the number of links, the StemID algorithm also calculates the change in transcriptome entropy. Differentiated cells usually express a small number of genes at high levels in order to perform cell specific functions, which is reflected by a low entropy. Stem cells and progenitor cells display a more diverse transcriptome reflected by high entropy (*Banerji et al., 2013*). By calculating the number of links of one cluster to other clusters and multiplying this with

the change in entropy, it generates a StemID score, which is representative to 'stemness' of a cell population.

Differential gene expression analysis was performed using the 'diffexpnb', which makes use of the DESeq algorithm. P-values were Benjamini-Hochberg corrected for false discovery rate to make the cutoff.

For the comparison between the embryonic and adult cardiomyocyte clusters, the number of differentially expressed genes between two clusters was calculated as described above and this was used as a measure of similarity between clusters.

## Inference of co-expressed gene modules

To identify modules of co-expressed genes along a specific differentiation trajectory (defined as a succession of significant links between clusters as identified by StemID) all cells assigned to these links were assembled in pseudo-temporal order based on their projection coordinate. Next, all genes that are not present with at least two transcripts in at least a single cell are discarded from the sub-sequent analysis. Subsequently, a local regression of the z-transformed expression profile for each gene is computed along the differentiation trajectory. These pseudo-temporal gene expression profiles are topologically ordered by computing a one-dimensional self-organizing map (SOM) with 1000 nodes. Due to the large number of nodes relative to the number of clustered profiles, similar profiles are assigned to the same node. Only nodes with more than three assigned profiles are retained for visualization of co-expressed gene modules. Neighboring nodes with average profiles exhibiting a Pearson's correlation coefficient >0.9 are merged to common gene expression modules. These modules are depicted in the final map. Analyses were performed as previously published (*Grün et al., 2016*).

Accession numbers mRNA-seq data are deposited on Gene Expression Omnibus, accession number GSE139218. Samples FK1 and FK2 represent adult cardiomyocytes. Samples LG-A and LG-B represent embryonic cardiomyocytes.

## Transmission Electron Microscopy

Hearts were excised and immediately chemically fixated at room temperature with 2,5% glutaraldehyde and 2% formaldehyde (EMS, Hainfield USA) in 0.1M phosphate buffer pH 7.4 for 2 hr. Next, hearts were post fixed with 1% $OsO_4$ (EMS, Hainfield USA)/1.5% $K_3Fe(CN)_6$ in 0.065 M phosphate buffer for 2 hr at 4°C and finally 1 hr with 0,5% uranyl acetate. After fixation, hearts were dehydrated in a graded series of acetone and embedded in Epon epoxy resin (Polysciences). Ultrathin sections of 60 nm were cut on a Leica Ultracut T (Leica, Vienna, Austria) and contrasted with uranyl acetate (0.4% in AD, EMS, Hainfield USA) and lead citrate (Leica Vienna, Austria) using the AC20 (Leica Vienna, Austria) and examined with a Jeol 1010 electron microscope (Jeol Europe, Nieuw Vennep, The Netherlands).

## Quantification of mitochondrial parameters

In every heart, each in the borderzone and remote myocardial region, 100 well-deliniated mitochondria with clearly visible outer and inner membranes were selected. Mitochondrial perimeter and surface were measured using the freehand tool of Image J. The perimeter to surface ratio was calculated and used as a factor that describes the pluriformity of mitochondria. The amount of cristae was estimated by counting the number of cristae intersected by a line of 0.5 µm length in 40 mitochondria per region.

## Histology and enzyme histochemistry

Serial cryosections of the heart were cut 7 µm thick and either fixed in formalin, stained with Meyer's hematoxylin and eosin (HE), dehydrated and mounted in Entellan, or incubated for enzyme histochemistry. Chemicals for histochemistry of succinate dehydrogenase (SDH) activity were obtained from Sigma Aldrich. Sections for SDH activity were incubated for 20 min at 28°C in 37.5 mM sodium phosphate buffer pH 7.60, 70 mM sodium succinate, 5 mM sodium azide and 0.4 mM tetranitro blue tetrazolium (TNBT). The reaction was stopped in 10 mM HCl. Controls without succinate did not stain. The incubated sections were mounted in glycerine gelatin. The absorbances of the SDH-

reaction product in the sections were determined at 660 nm using a calibrated microdensitometer and ImageJ.

## Pharmacological inhibition of glycolysis

ZEBRAFISH: Zebrafish were injured and received intraperitoneal (i.p.) injections twice daily with either PBS or 2-Deoxy-D-Glucose (Sigma-Aldrich, 1 mg/g) from days 3 to 6 and one more injection on day seven after injury, two hours before fish were euthanized and hearts harvested. I.p. injections were performed using a Hamilton Syringe (gauge 30) as described in literature (*Kinkel et al., 2010*). Injection volumes were corrected to body weight (30 µl/g).

MAMMALIAN P7 CARDIAC CULTURES: Primary cardiac cultures were isolated from P7 mice using a neonatal dissociation kit (Miltenyi Biotec,130-098-373) using the gentleMACS homogenizer, according to the manufacturer's instructions and cultured in Gelatin-coated (0.1%, G1393, Sigma) wells with DMEM/F12 medium supplemented with L-glutamine, Na-pyruvate, nonessential amino acids, penicillin, streptomycin, 5% horse serum and 10% FBS ('complete-medium') at 37∘C and 5% $CO_2$ for 24 hr. Afterwards, medium was replaced with FBS-depleted medium (otherwise same composition) for additional 48 hr of culture in either 3 mM 2DG (Sigma-Aldrich) or 80 µM lonidamine (Sigma-Aldrich) before further processing.

## Ex vivo glucose uptake

Fish were euthanized on ice water before hearts were extracted in PBS + heparin and were allowed to bleed out for 15 min. Hearts were then transferred into fresh PBS + 10%KCl to stop the heart from beating and mounted directly on a glass bottom cell culture dish in 1% agarose. Thereafter, 2NBDG (Caymanchem #11046, 400 µM) was added to the dish and the hearts were taken directly for imaging. Imaging was performed using a Leica SP5 multiphoton microscopy using 930 nm laser excitation wavelength. 150 µm z-stacks were made with z-step size 5 µm every 5 min for 2 hr.

## Gene set enrichment analysis (GSEA)

GSEA (Genepattern, Broad Institute) was performed to assess enrichment for glycolytic genes upregulated between cluster 7 and 2. A list of genes involved in zebrafish glycolysis was obtained from KEGG. As number of permutations 1000 was used, which means $p=0$ indicates $p<0.001$.

## Statistical analysis of data

All statistical testing was performed by unpaired T-tests besides zebrafish qPCR data for which a one-way ANOVA was performed.

# Acknowledgements

We would like to thank V Christoffels for critical reading of the manuscript and Life Science Editors for editing support.

# Additional information

## Funding

| Funder | Grant reference number | Author |
| --- | --- | --- |
| ERA-CVD | JCT2016-40-080 | Gilbert Weidinger<br>Eldad Tzahor<br>Jeroen Bakkers |
| Deutsche Forschungsgemeinschaft | 251293561 | Gilbert Weidinger |
| Netherlands Heart Foundation NHS/CVON | Cobra3 | Jeroen Bakkers |
| European Molecular Biology Organization | ALTF1129-2015 | Phong D Nguyen |
| Human Frontier Science Program | LT001404/2017-L | Phong D Nguyen |

| Nederlandse Organisatie voor Wetenschappelijk Onderzoek | 016.186.017-3 | Phong D Nguyen |
| --- | --- | --- |
| Deutsche Forschungsgemeinschaft | 316249678 | Gilbert Weidinger |
| Deutsche Forschungsgemeinschaft | 414077062 | Gilbert Weidinger |
| NIH Clinical Center | RO1 HL081674 | Kenneth D Poss |
| NIH Clinical Center | R01 HL131319 | Kenneth D Poss |
| NIH Clinical Center | R01 HL136182 | Kenneth D Poss |
| ERC | StG281289 | Eldad Tzahor |
| ERC | AdG788194 | Eldad Tzahor |
| Fondation Leducq Transatlantic Network of Excellence | 15CVD03 | Eldad Tzahor |

The funders had no role in study design, data collection and interpretation, or the decision to submit the work for publication.

## Author contributions

Hessel Honkoop, Conceptualization, Data curation, Formal analysis, Investigation, Visualization, Methodology, Writing - original draft; Dennis EM de Bakker, Conceptualization, Data curation, Formal analysis, Investigation, Visualization, Methodology, Assisted with zebrafish experiments, Writing - review and editing; Alla Aharonov, Conceptualization, Data curation, Formal analysis, Investigation, Visualization, Methodology, Writing - review and editing; Fabian Kruse, Conceptualization, Data curation, Formal analysis, Investigation, Methodology, Writing - original draft; Avraham Shakked, Data curation, Formal analysis, Assisted with the mouse experiments; Phong D Nguyen, Conceptualization, Formal analysis, Investigation, Writing - review and editing; Cecilia de Heus, Data curation, Formal analysis, Visualization, Methodology, Performed the EM experiments, Writing - review and editing; Laurence Garric, Data curation, Formal analysis, Investigation, Methodology, Assisted with scRNAseq experiments; Mauro J Muraro, Conceptualization, Data curation, Formal analysis, Visualization, Methodology, Assisted with scRNAseq experiments; Adam Shoffner, Data curation, Formal analysis, Methodology, Performed the Nrg1 OE zebrafish experiments; Federico Tessadori, Conceptualization, Formal analysis, Supervision, Methodology, Generated the tg(nppaBAC:mCitrine) zebrafish line, Writing - review and editing; Joshua Craiger Peterson, Data curation, Formal analysis, Supervision, Investigation, Methodology, Assisted with the tg(nppaBAC:mCitrine) zebrafish line; Wendy Noort, Formal analysis, Investigation, Assisted with SDH experiment; Alberto Bertozzi, Formal analysis, Investigation, Methodology, Assisted with zebrafish cryoinjury experiments; Gilbert Weidinger, Conceptualization, Supervision, Assisted with zebrafish cryoinjury experiments; George Posthuma, Conceptualization, Formal analysis, Supervision, Methodology, Assisted with EM experiments; Dominic Grün, Conceptualization, Formal analysis, Methodology, Assisted with scRNAseq data analysis; Willem J van der Laarse, Supervision, Methodology, Assisted with SDH experiment; Judith Klumperman, Supervision, Funding acquisition, Methodology, Assisted with EM experiment; Richard T Jaspers, Conceptualization, Formal analysis, Supervision, Methodology, Assisted with SDH experiment; Kenneth D Poss, Conceptualization, Supervision, Funding acquisition, Assisted with Nrg1 OE zebrafish experiment; Alexander van Oudenaarden, Conceptualization, Supervision, Funding acquisition, Assisted with scRNAseq data analysis; Eldad Tzahor, Conceptualization, Supervision, Funding acquisition, Writing - review and editing; Jeroen Bakkers, Conceptualization, Formal analysis, Supervision, Funding acquisition, Visualization, Project administration, Writing - original draft

## Author ORCIDs

Gilbert Weidinger https://orcid.org/0000-0003-3599-6760
Richard T Jaspers https://orcid.org/0000-0002-6951-0952
Eldad Tzahor https://orcid.org/0000-0002-5212-9426
Jeroen Bakkers https://orcid.org/0000-0002-9418-0422

## Ethics

Animal experimentation: All experiments were conducted in accordance with the ethical guidelines. Animal experiments were approved by the Institutional Animal Care and Use Committee (IACUC) of the Royal Dutch Academy of Sciences (AVD801002016404), the state of Baden-Württemberg and the animal protection representative of Ulm University (Tierversuch 1352), Duke University (A057-18-02) and the Weizmann Institute (13240419-3).

## Decision letter and Author response

Decision letter https://doi.org/10.7554/eLife.50163.sa1
Author response https://doi.org/10.7554/eLife.50163.sa2

# Additional files

## Supplementary files

• Transparent reporting form

## Data availability

Sequencing data have been deposited in GEO under accession code GSE139218. Other data generated during this study are included in the manuscript and supporting files.

The following dataset was generated:

| Author(s) | Year | Dataset title | Dataset URL | Database and Identifier |
| --- | --- | --- | --- | --- |
| Honkoop H, de Bakker DE, Aharonov A, Kruse F, Shakked A, Nguyen PD, de Heus C, Garric L, Muraro MJ, Shoffner A, Tessadori F, Peterson JC, Noort W, Bertozzi A, Weidinger G, Posthuma G, Grün D, van der Laarse WJ, Klumperman J, Jaspers RT, Poss KD, van Oudenaarden A, Tzahor E, Bakkers J | 2019 | Single-cell analysis uncovers that metabolic reprogramming is essential for cardiomyocyte proliferation in the regenerating heart. | http://www.ncbi.nlm.nih.gov/geo/query/acc.cgi?acc=GSE139218 | NCBI Gene Expression Omnibus, GSE139218 |

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
