## [Decision Letter]

**Acceptance summary:**

This is a strong and well-written paper with fascinating observations on the roles of the glycolytic pathway in regenerating hearts, Erbb2 upregulation, glycolytic pathway upregulation and cardiomyocyte proliferation. The results indicate that ErbB2 signaling in cardiomyoctes drives a switch from OXPHOS to glycolytic metabolism, resulting in reentry into the cell cycle during the regeneration response.

**Decision letter after peer review:**

Thank you for submitting your article "Single-cell analysis uncovers metabolic reprogramming in proliferating cardiomyocytes of the regenerating heart" for consideration by *eLife*. Your article has been reviewed by three peer reviewers, and the evaluation has been overseen by Marianne Bronner as the Senior and Reviewing Editor. The following individual involved in review of your submission has agreed to reveal their identity: Shawn Burgess (Reviewer #1).

The reviewers have discussed the reviews with one another and the Reviewing Editor has drafted this decision to help you prepare a revised submission.

Summary:

This paper leads with very useful single cell transcriptome analysis of adult zebrafish hearts in response to injury, compared to developing hearts in zebrafish embryos, with a general theme that regeneration response reprograms cells toward dedifferentiation and proliferation. The combination of a transgenic line that labels cardiomyocytes at the 'border zone' near the injury site and ssRNA-seq reveals important local changes in metabolic genes, and metabolic activity, in response to cardiac injury, correlated with increases in proliferation. This concurs with an interesting change in mitochondria morphology. From these results, the focus is turned to the roles of the glycolytic pathway in regenerating hearts, and pathway of Erbb2 upregulation, glycolytic pathway upregulation and cardiomyocyte proliferation. The results indicate that ErbB2 signaling in cardiomyoctes drives a switch from OXPHOS to glycolytic metabolism, resulting in rentry into the cell cycle. Overexpression of ErbB2 in mice hearts gives a similar switch in metabolic pathway gene expression. Inhibition of glycolysis by 2-Deoxy glucose (2DG) treatments reduces the proportion of PCNA-labeled cardiomyocytes.

Essential revisions:

The full reviews below provide some guidance about revisions that are textual or require checks of statistical analysis and cut-off parameters.

Reviewer #1:

Honkoop and colleagues have submitted a manuscript describing the isolation and scRNA-seq experiments performed on regenerating adult zebrafish hearts. Specifically they used a *nppa*:mCitrine transgenic reporter to FAC sort cells from the "border zone" of the regenerating heart. Bioinformatic analysis demonstrated a de-differentiation of the cardiomyocytes towards a more embryonic-like gene state and that this transition was triggered by ErbB2 signaling. They also showed an obligate metabolic shift from aerobic to anaerobic respiration triggered by the ErbB2 signals and that regeneration was significantly inhibited if glycolysis was blocked by 2DG. Despite a more limited ability to regenerate in mice, they were able to also show similar effects in mouse cardiomyocytes, particularly in a transgenic model expressing constitutively active form of ErbB2. These results are interesting, well controlled and of general interest to the readers of *eLife*.

Reviewer #2:

1) The authors are commendably careful in their title, summary and discussion of the results with respect to cardiac regeneration. They have not shown that inhibition of the glycolytic pathway alters the ability of the zebrafish heart to regenerate, only that it reduces reentry into the cell cycle, which of course is a step used in regeneration. The single cell transcriptome analysis and most of the gene expression analysis was done at 7 days post injury (dpi), which is appropriate to capture the early steps in the process, during the cardiomyocyte responses to heart injury. In contrast, to test the effects of functional alteration of metabolic pathways on regeneration per se, the hearts would have to be analyzed later in the process, around 30 days post injury, which has not been provided. In keeping with this, one sentence at the end of the summary "Together these results reveal a new mechanism in which glycolysis regulates cardiomyocyte proliferation and heart regeneration" should be changed to "during heart regeneration."

2) In Figure 6, the Nrg1 OE hearts look quite different morphologically from the control hearts. Please discuss what is happening; is the overexpression significantly altering the heart function/physiology? It is also interesting that Nrg1/caErbB2 in fish/mice hearts appears to induce overexpression of glycolytic genes in some parts of the heart, but not others, perhaps compact outer myocardium in fish and inner myocardium in mice. Is this observation consistent? If so, worth a brief mention.

3) In Figure 6, it appears that in the PCR analysis, HK1 increased expression in response to injury (as seen in situ, Figure 4) and in Nrg1 OE hearts, but does not appear to be decreased by inhibition of Erbb2. This is worth a brief comment.

4) In Figure 5, it would be useful to see a larger view of the hearts, not just the zoomed in region of the border zone.

Reviewer #3:

Here, Honkoop et al. generated a transgenic line that labels proliferating cardiomyocytes at the wound border zone. Using this line, the authors isolate border zone cardiomyocytes using SORT-seq and find that border zone cardiomyocytes have a distinct transcriptional profile that resembles embryonic cardiomyocytes. In this regard, the border zone cardiomyocytes have an upregulation of glycolysis gene expression suggesting a metabolic reprogramming is required for proliferation of cardiomyocytes at the onset of injury. Furthermore, the authors describe a mechanism underlying metabolic reprogramming by induction of Nrg1/ErbB2 signaling. Overall, the authors contribute to our understanding of how regeneration reactivates developmental programs by activating a transcriptional profile similar to embryonic populations including metabolic reprogramming to a more embryonic mechanism.

1) The authors analyzed 352 cells after a cutoff of a minimum of 3500 reads per cells. This seems a bit high of a cutoff, and with such stringency, they could be missing heterogeneity within cardiomyocytes populations by analyzing roughly half of the cells originally collected. Could the authors comment on why they decided on such a high reads per cell cutoff?

2) A GO analysis of the different single cell clusters would highlight the similarities and differences between clusters. It would be particularly interesting to see GO term analyses when comparing the embryonic and proliferating cardiomyocyte cluster.

3) Pseudotime analysis suggested that glycolysis gene induction preceded embryonic gene expression. Is this known to be the same order of events in the developing heart?

---

## [Author Response]

Essential revisions:Reviewer #2:1) The authors are commendably careful in their title, summary and discussion of the results with respect to cardiac regeneration. They have not shown that inhibition of the glycolytic pathway alters the ability of the zebrafish heart to regenerate, only that it reduces reentry into the cell cycle, which of course is a step used in regeneration. The single cell transcriptome analysis and most of the gene expression analysis was done at 7 days post injury (dpi), which is appropriate to capture the early steps in the process, during the cardiomyocyte responses to heart injury. In contrast, to test the effects of functional alteration of metabolic pathways on regeneration per se, the hearts would have to be analyzed later in the process, around 30 days post injury, which has not been provided. In keeping with this, one sentence at the end of the summary "Together these results reveal a new mechanism in which glycolysis regulates cardiomyocyte proliferation and heart regeneration" should be changed to "during heart regeneration."

We agree that the research performed in this manuscript is limited to the cardiomyocyte proliferation aspect of zebrafish heart regeneration and have changed the sentence at the end of the summary as suggested.

2) In Figure 6, the Nrg1 OE hearts look quite different morphologically from the control hearts. Please discuss what is happening; is the overexpression significantly altering the heart function/physiology?

As a response to the Nrg1 overexpression, the cardiomyocytes in the cortical layer will disassemble sarcomere structures and initiate cell proliferation, resulting in a thickening of mainly the cortical layer, which has been described in Gemberling et al., 2015. The overexpression of *nrg1* leads to a specific and consistent thickening of the ventricular wall already at 14dpt. Further analysis on the model revealed this thickening was accomplished by an increase in cardiomyocyte number rather than hypertrophy of existing cardiomyocytes. At first, physiology in *nrg1* OE fish is unaffected, but after three months it appears cardiac function improves as a result of the overexpression as atrioventricular filling improves. This beneficial effect turns into a negative effect after 8 months of overexpression, due to deleterious growth.

It is also interesting that Nrg1/caErbB2 in fish/mice hearts appears to induce overexpression of glycolytic genes in some parts of the heart, but not others, perhaps compact outer myocardium in fish and inner myocardium in mice. Is this observation consistent? If so, worth a brief mention.

The localization of glycolysis markers is consistent between biological repeats, in both the zebrafish and mouse hearts.

In the zebrafish the tested glycolysis markers show the strongest signal in the cortical layer coinciding with high proliferation in this region. It is still unknown why the effect of *nrg1* OE in this model leads to a more pronounced effect in the cortical myocardium compared to the trabecular myocardium.

We have made some textual changes to clarify the effect of *nrg1* OE in the zebrafish heart: “In this model especially cortical cardiomyocytes start to divide after tamoxifen injection leading to thickening of this layer. We observed a profound and consistent upregulation of glycolysis genes in the cortical myocardium coinciding with the reported cardiomyocyte dedifferentiation and proliferation in this layer (Figure 6A).”

In the caErbB2 OE mouse model we observed consistent pPDH signal in the inner myocardial layer, which we now mention in the subsection “Activating ErbB2 signaling induces a metabolic switch from OXPHOS to glycolysis and lactate fermentation in murine cardiomyocytes”. We don’t have a good explanation for the stronger pPDH staining in the inner layer compared to the outer layer. In the caErbB2 OE model cardiomyocyte proliferation occurs throughout the myocardium.

3) In Figure 6, it appears that in the PCR analysis, HK1 increased expression in response to injury (as seen in situ, Figure 4) and in Nrg1 OE hearts, but does not appear to be decreased by inhibition of Erbb2. This is worth a brief comment.

The reviewer rightly points to the fact that *hk1* is not downregulated in the border zone after ErbB2 inhibition. On the contrary, we have shown that Nrg1 is indeed able to upregulate *hk1* expression in the Nrg1 OE model. This discrepancy can most likely be explained by redundant upstream factors accounting for border zone *hk1* expression in the absence of Nrg1/ErbB2 signaling. We have added a brief comment on this in the revised manuscript: “In contrast to the other glycolysis genes, *hk1* expression was not reduced after AG1478 treatment likely as a result of redundant signaling pathways in the borderzone.”.

4) In Figure 5, it would be useful to see a larger view of the hearts, not just the zoomed in region of the border zone.

A larger view of the hearts treated either with PBS or 2DG has been added to Figure 5 to give a better overview of the hearts 7 days post injury.

Reviewer #3:[…] 1) The authors analyzed 352 cells after a cutoff of a minimum of 3500 reads per cells. This seems a bit high of a cutoff, and with such stringency, they could be missing heterogeneity within cardiomyocytes populations by analyzing roughly half of the cells originally collected. Could the authors comment on why they decided on such a high reads per cell cutoff?

Here, the reviewer raises a valid question on how we determined the read cutoff for our single cell sequencing analysis. We had a dataset consisting of two different libraries of 384 cells each for a combined dataset of 768 cells. In the combined library we detected an average of 10,443 reads per cell. Based on the distribution of the log10 total reads plotted against the frequency, we decided to introduce a cutoff at minimally 3500 reads per cell before further analysis (see diagnostic graph in Author response image 1). This reduced the number of cells to 352. Cutoffs depend on cell type and hence can vary between different organs with different cell type composition. Importantly, lowering the cutoff to a lower number of reads will result in more included cells but will result in the loss of information on lowly expressed transcript as a result of down sampling of the data to the read cutoff. In this process reads in a cell are lost in order to make cells with different read counts comparable. With the chosen cutoff we lost the minimum of rare transcript information while including enough cells to not miss any heterogeneity within the cardiomyocyte population.

In order to justify the choice of thresholds for filtering we performed the analysis at both lower and higher cutoffs. These analyses show that while the amount of included cells is influenced by different cutoffs, the clustering pattern is retained (Author response image 1). This is also recapitulated by the expression of two example genes *nppa* and *ttna*, both highly expressed in the cells which are referred to as cluster 7 in the original manuscript (Author response image 1). Note that the number of clusters decreases when a lower read cutoff is used as a result of down sampling (n=5 for cutoff of 500, n=12 for cutoff of 3500), leading to the loss of valuable information.

**Author response image 1. respfig1:** Different read cut-offs do not alter clustering pattern, while altering number of included cells. (**A**) Distribution of reads per cell. Green line represents 3,500 reads, red line indicates the mean number of reads per cell (10,443). (**B**) tSNE maps generated with different read cutoffs. Read cutoffs and included number of cells are indicated on top of tSNE maps. (**C**) Expression of two marker genes of dedifferentiated cardiomyocytes: *nppa* and *ttna*.

2) A GO analysis of the different single cell clusters would highlight the similarities and differences between clusters. It would be particularly interesting to see GO term analyses when comparing the embryonic and proliferating cardiomyocyte cluster.

We agree that performing GO term analysis on our single cell data could give valuable insights into processes that differ between the clusters. To gain more insight into this DAVID functional annotation was used to get info on these mechanisms.

First, we performed GO term analysis on adult cardiomyocytes only. Here we compared the dedifferentiated cardiomyocytes from cluster 7 to the more mature cardiomyocytes in cluster 2 (Author response image 2). As expected from pseudo time analysis we found multiple GO terms related to metabolism differentially regulated between clusters. Moreover, translation and sarcomere related GO-terms were upregulated in cluster 7.

Next, we compared the proliferating adult cardiomyocyte cluster 5 from our combined analysis to embryonic cardiomyocyte cluster 6. This analysis suggests that proliferating adult cardiomyocytes are highly active in sarcomere assembly and have more oxidative capacity than embryonic cardiomyocytes (or are in the process of establishing this). Embryonic cardiomyocytes on the other hand expressed genes involved in heart development, production of extracellular matrix and developmental pathways such as Wnt and Tgf-b.

We have added a complete list of GO-terms for the comparison of proliferating (cluster 7) and non-proliferating (cluster 2) adult cardiomyocytes to Figure 1—source data 3 as well as a smaller list of GO-terms to Figure 4—figure supplement 1. Moreover, we have added the complete lists of GO-terms comparing all embryonic and adult cardiomyocyte clusters from our combined analysis to Figure 2—source data 4.

**Author response image 2. respfig2:** GO-term analysis on differentially expressed genes in cardiomyocyte clusters. Representative GO-terms for cardiomyocyte clusters. GO-terms in upper table are obtained from differentially expressed genes between adult clusters 2 and 7 (p<0.01). Lower table GO-terms are obtained from differentially expressed genes between cluster 5 (Proliferating adult CM) and cluster 6 (Embryonic CM) of the combined single-cell analysis.

3) Pseudotime analysis suggested that glycolysis gene induction preceded embryonic gene expression. Is this known to be the same order of events in the developing heart?

This is an interesting question. Glycolysis genes such as *eno1a, hk1* and *ldha* are already well expressed in the gastrulating (shield stage) embryo (based on tomo-seq data: http://zebrafish.genomes.nl). In addition, we have performed in situhybridizations for these glycolytic genes in later stages of the developing embryo and observed high expression in the developing embryo including the heart (Author response image 3). Interestingly, glycolysis genes are highly expressed in the developing heart at 28 and 48 hpf, but much lower at later stages of development (5dpf). This observation suggests that there is an inverse correlation of glycolytic gene expression and the maturation state of cardiomyocytes, which is in good agreement with data from other animal models.

**Author response image 3. respfig3:** Glycolytic gene expression in the embryonic zebrafish heart. Expression of *eno1a* (**a, b, c, d**), *hk1* (**e, f, g, h**) and *ldha* (**i, j, k, l**) during zebrafish development at 28hpf (**a, e, i**), 48 hpf (**b, f, j**) and 5dpf (**c, d, g, h, k, l**) respectively. Outline of the heart is indicated by dashed lines.